# HQ-Edit: A High-Quality Dataset for Instruction Based Image Editing

**Mude Hui**[1][†]**, Siwei Yang**[1][†]**, Bingchen Zhao**[2]**, Yichun Shi**[3]**, Heng Wang**[3]**,
Peng Wang**[3]**, Cihang Xie**[1]**,Yuyin Zhou**[1]

[†]equal contribution

[1]University of California, Santa Cruz       [2]University of Edinburgh       [3]ByteDance

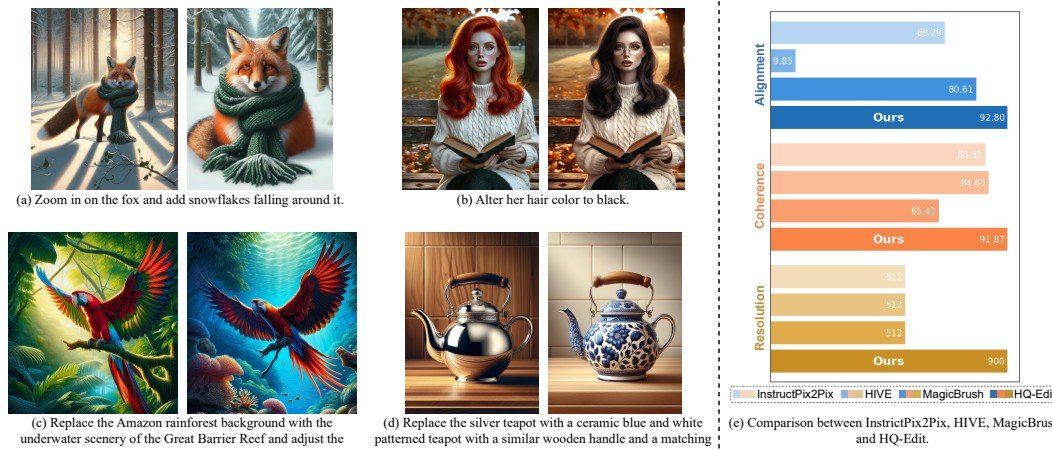

(a) Zoom in on the fox and add snowflakes falling around it.

(b) Alter her hair color to black.

(c) Replace the Amazon rainforest background with the underwater scenery of the Great Barrier Reef and adjust the parrot's position and wings to depict it flying.

(d) Replace the silver teapot with a ceramic blue and white patterned teapot with a similar wooden handle and a matching ceramic lid.

(e) Comparison between InstrictPix2Pix, HIVE, MagicBrush and HQ-Edit.

Figure 1: (a) - (d): example images and edit instructions from HQ-Edit. (e): we compare the dataset quality between our HQ-Edit and existing ones. Note that "Alignment" and "Coherence" are our newly developed metrics (introduced in Sec. 3.4) for measuring image/text qualities.

## Abstract

This study introduces HQ-Edit, a high-quality instruction-based image editing dataset with around 200,000 edits. Unlike prior approaches relying on attribute guidance or human feedback on building datasets, we devise a scalable data collection pipeline leveraging advanced foundation models, namely GPT-4V and DALL-E 3. To ensure its high quality, diverse examples are first collected online, expanded, and then used to create high-quality diptychs featuring input and output images with detailed text prompts, followed by precise alignment ensured through post-processing. In addition, we propose two evaluation metrics, Alignment and Coherence, to quantitatively assess the quality of image edit pairs using GPT-4V. HQ-Edit's high-resolution images, rich in detail and accompanied by comprehensive editing prompts, substantially enhance the capabilities of existing image editing models. For example, an HQ-Edit finetuned InstructPix2Pix can attain state-of-the-art image editing performance, even surpassing those models fine-tuned with human-annotated data.

## 1 Introduction

The recent advancements in text-to-image generative models (Rombach et al., 2022; Ramesh et al., 2022; Gu et al., 2022; Saharia et al., 2022; Huang et al., 2024) have catalyzed a new era in diverse real-world applications ranging from advertising and photography to digital art and movie production. Among these generative models, applications of domain-specific image conditioned generations (Ruiz et al., 2023; Ye et al., 2023; Wang & Shi, 2023; Hu et al., 2023), and multi-modal non-specific generation methods (Pan et al., 2023; Sheynin et al., 2023; Wu et al., 2023) have gathered significant attention.

Our work concentrates on applications of highly accurate, general instruction-based single image editing without relying on external attribute guidance, as proposed in previous studies (Avrahami et al., 2022; Hertz et al., 2022; Ling et al., 2021; Wallace et al., 2023; Shi et al., 2022). We identify that this particular challenge has not been adequately addressed in the literature yet. To the best of our knowledge, one of the major hurdles in training an instruct-based image editing model lies in the limited availability of high-quality datasets pairing editing instructions with corresponding images. This challenge was best tackled by the seminal work InstructPix2Pix (Brooks et al., 2023). Specifically, it first leverages GPT-3 (Brown et al., 2020) to generate both an instruction and an edited image caption based on a given image description; then, it applies Stable Diffusion (SD1.5) (Rombach et al., 2022) and Prompt-to-Prompt (Hertz et al., 2022) to create the paired input and output images. However, their underlying models, namely SD1.5 and GPT-3, are outdated compared to current state-of-the-art counterparts such as DALL-E 3 and GPT-4. Consequently, these models produce images with lower resolution and suboptimal edit-image alignment. Subsequent studies also attempted to improve it via incorporating human feedback (Zhang et al., 2023) or segmentation masks (Chakrabarty et al., 2023; Zhang et al., 2024), yet the generated data continue to exhibit one or more of the aforementioned issues, as showcased in Figure 1.

In this work, we aim to leverage the ability from the best text-image models, *i.e.*, DALL-E 3 (OpenAI, 2023a), GPT4 & GPT4V (OpenAI, 2023b), to build a *high-quality* dataset for improving the image editing datasets. Ideally, in case of accessing the model weights, it should provide high-resolution images that offer rich detail, both in their visual content and the accompanying instructions; Also, it should provide more precise alignment between textual instructions and image pairs, ensuring edits are applied as directed while maintaining fidelity in areas not subject to modification.

However, only with the access to their APIs, in this study, we discover a way of pair image generation with DALL-E 3 based on prompt-engineer, which enable a similar Prompt-to-Prompt process, yielding high-quality editing image pairs, which we name as **HQ-Edit**. HQ-Edit provides a significant leap forward, featuring high image resolutions of approximately $900 \times 900$ pixels—nearly double that of existing datasets, and comprises around 200,000 detailed edit instructions. Moreover, unlike prior approaches relying on attribute guidance or human feedback, HQ-Edit is synthetically generated through a scalable pipeline that harnesses the image text understanding capabilities of powerful foundation models of GPT-4V and DALL-E 3.

Our data curation process comprises three key steps: **Expansion** - **Generation** - **Post-processing**. Firstly, in the *Expansion* phase, we extract seed triplets with high diversity—consisting of input/output image descriptions along with edit instructions—from online sources. Subsequently, we leverage GPT-4 to expand these initial triplets into around 100,000 instances, ensuring the comprehensive diversity of edit instructions. In the subsequent *Generation* phase, the seed triplets are processed by GPT-4 to merge and refine into detailed diptych prompts for DALL-E 3, creating diptychs with input and output image pairs displayed side-by-side. Note this diptych-based prompting design is motivated by the finding that, compared to generating input images and output images separately, generating diptychs generally exhibits superior quality, with better alignment and consistency in edit-irrelevant areas. Lastly, the generated diptychs and refined prompts undergo *post-processing* to ensure precise alignment between the paired images and their corresponding instructions. Specifically, 1) each diptych is decomposed into paired images, which undergo warping and filtering to ensure correspondence; 2) the instructions are refined using rewritten instructions from GPT-4V; and 3) the inverse-edit instructions are also generated, allowing for the transformation of output images back into their input counterparts.

On top of HQ-Edit, we introduce two metrics, **Alignment** and **Coherence**, to comprehensively and quantitatively evaluate the quality of image edit pairs. The first metric, *Alignment*, checks for semantic consistency with the edit prompt, ensuring accurate modification of mentioned objects while preserving image fidelity. The second metric, *Coherence*, evaluates the edited image's aesthetic quality, including lighting and shadow consistency, style coherence, and edge smoothness. Extensive empirical results show that our synthetically created HQ-Edit can even surpass human-annotated data in enhancing instruction-based image editing models. For example, the HQ-Edit finetuned InstructPix2Pix model substantially outperforms its vanilla version, achieving a 12.3 increase at Alignment, and a 5.64 enhancement at Coherence.

## 2 RELATED WORKS

**Text Guided Image Editing Model** Text guided image editing models have been extensively discussed recently. Prompt2Prompt (Hertz et al., 2022) modifies words in the original prompts to perform both local editing and global editing by cross-attention control. Imagic (Kawar et al., 2023) optimizes a text embedding that aligns with the input image, then interpolates it with the target description, thus generating correspondingly different images for editing. DiffEdit (Couairon et al., 2022) locate edit position based on text (generate mask), and limit diffusion model to generate the mask area. An important type of Text Guided is the instruction, which describes where, what and how an image should be edited. Instruction-based image editing model will follow the instruction without requiring elaborate descriptions or region masking, and enables users to modify images more easily and flexibly. InstructPix2Pix (Brooks et al., 2023) is the first instruction-based image editing model, by fine-tuning the Stable Diffusion (Rombach et al., 2022) on a dataset of image editing examples, which generated by GPT-3 (Brown et al., 2020) and Prompt2Prompt. Subsequent work, such as HIVE (Zhang et al., 2023) and Magicbrush (Zhang et al., 2024), have focused on improving the quality or quantity of the dataset.

**Instruction-based Image Editing Datasets** Since collecting high-quality open data for image editing can be challenging, early approaches construct datasets by manually labeling image pairs (Zhang et al., 2024). While this ensured a degree of quality, it inherently restricted the scale and diversity of the dataset. For example, Magicbrush (Zhang et al., 2024) contains about only 10,000 edits, and predominantly focuses on object-level transformations, largely overlooking global edits like style or weather changes. On the other hand, there have been endeavors to synthesize large-scale datasets. For example, InstructPix2Pix (Brooks et al., 2023) leverages GPT-3 and Prompt2Prompt (Hertz et al., 2022) to generate editing pairs, and HIVE (Zhang et al., 2023) introduces reinforcement learning from human feedback to align the data with human expectations. However, these synthetic data often have the drawback of low quality and inaccurate editing, resulting in such trained image editing models outputting low-quality images and deviating from the actual edit instructions. FaithfulEdits (Chakrabarty et al., 2023) attempts to mitigate these issues by using inpainting techniques, followed by a filtering process involving VQA models. Yet, this method tends to underperform, particularly in global edits requiring extensive image modification, like style transfer.

Different from existing approaches, we leverage the latest foundation models, GPT-4 and DALL-E 3, to generate high-quality image editing pairs at scale. We also introduce additional enhancements, *e.g.*, using GPT-4V to rewrite the edit instruction to align with the images more closely.

## 3 HQ-EDIT DATASET

The process of collecting HQ-Edit, illustrated in Figure 2, comprises three phases. Initially, triples of input/output image descriptions and edit instructions are expanded into 100,000 instances during the Expansion phase (Section 3.1). Subsequently, these instances are refined into detailed prompts for DALL-E 3 to generate diptychs in the Generation phase (Section 3.2). Finally, alignment and refinement occur in the Post-processing phase (Section 3.3).

### 3.1 EXPANSION

As in Figure 2, we first collect a small yet representative dataset comprising 203 samples from online sources as the seed. To ensure alignment between the text descriptions and image pairs, we manually revise the descriptions based on the disparities in content. Additionally, we include 90 samples from the Emu Edit (Sheynin et al., 2023) test set. We refer to these 293 samples as seed triplets, with each triplet comprising input/output image descriptions along with corresponding edit instructions.

To increase its size, we follow the pipeline in Self-instruct (Wang et al., 2022), which applies large language models on a small set of seed samples to generate a large volume of expansions that are both high in quality and consistent with the seed structure. Specifically, we utilize GPT-4 to expand this initial set of 293 seed triplets into around 100,000 instances, ensuring a thorough representation of diverse image editing scenarios. This strategy not only broadens the scope of edit instructions but also leverages GPT-4's knowledge to enrich the diversity and detail of image descriptions and edit instructions.

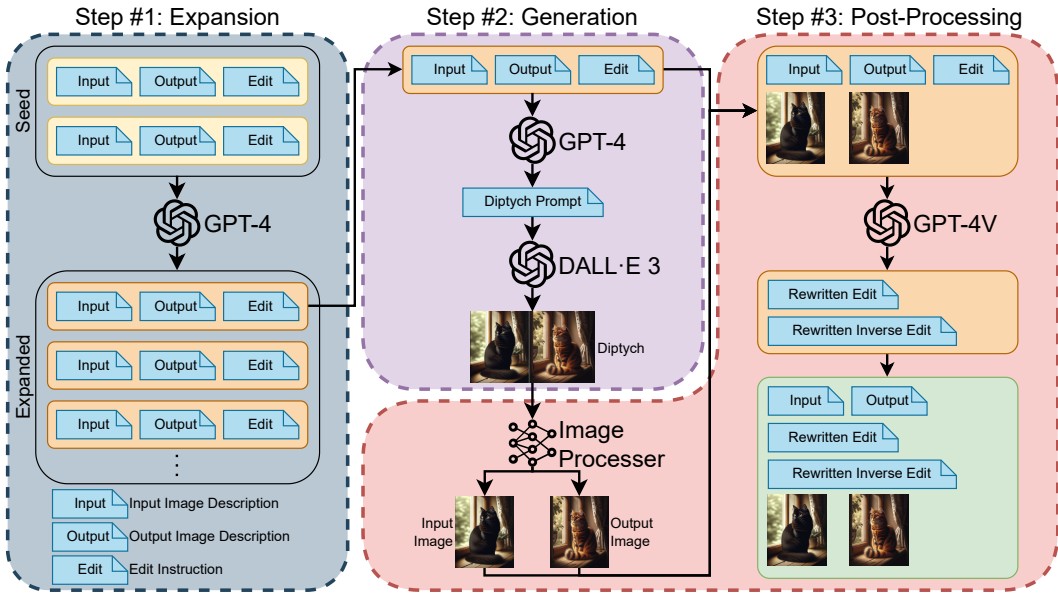

Figure 2: Our method consists of three steps: (1) Expansion: Massively generating image descriptions and edit instructions based on seed samples using GPT-4. (2) Generation: Generating diptychs using GPT-4V and DALL-E according to image descriptions and instructions. (3) Post-Processing: Post-process diptychs and edit instructions with GPT-4V and other various methods to produce image pairs and further enhance the quality of the dataset in different aspects.

Table 1: An example of the diptych prompt.

| Input/Output/Edit | Diptych Prompt For DALL-E 3 |
| --- | --- |
| **Input:** a graffiti-covered urban alley | Generate a diptych with two side-by-side images. On the left, depict a vibrant, narrow urban alley teeming with colorful graffiti on its walls. Details should include assorted tags and street art in various styles, with a depth indicating the alley stretches far back. Miscellaneous urban elements like a dumpster, a stray cat, and fire escape ladders should be present, and a subtle sunlight to cast soft shadows, indicating a daytime setting. On the right, replicate this scene exactly but convert the image into high-contrast black and white with stark lighting to enhance textures and shadows, and accentuate the details of the graffiti, giving an edgy, gritty aesthetic. Each element from the left image must be recognizable in monochrome, especially the contrasts between the shaded areas and the illuminated ones created by an overhead midday light. |
| **Edit:** present the photo with a high-contrast black and white effect | |
| **Output:** a high-contrast black and white image of a graffiti-covered alley | |

## 3.2 GENERATION

Upon acquiring the essential instructions and image descriptions from Expansion, the next step is to generate paired images that align with the instruction data. We hereby employ DALL-E 3 (OpenAI, 2023a), a state-of-the-art image generation model capable of producing high-resolution images based on textual descriptions. However, DALL-E 3 is not originally designed for instruction-based image editing, and therefore cannot directly produce paired images. Thus, we devised a workaround by creating diptychs consisting of input and output images side by side, followed by post-processing (Section 3.3) to reconstruct paired images. Interestingly, we note that generating input and output images together in diptych form, rather than separately, significantly enhances the relevance and correspondence between image pairs. As outlined in Figure 2, each triplet is fed to GPT-4 to form a diptych prompt for DALL-E 3 to generate a diptych. Moreover, to refine the diptych prompts and improve consistency between image pairs, GPT-4 is also utilized to elaborate further on the prompts. For instance, a basic description like *"an elder Asian woman"* can be enriched into *"an elderly East Asian woman with wrinkle-lined skin and white hair pulled back neatly, wearing a traditional gold silk hanbok"*. This enrichment adds complexity to the prompts and subsequently to the generated diptychs. An example of the enhanced diptych prompt is shown in Table 1. Overall, this process yields 98,675 data samples comprising input-output text pairs, edit instructions, and diptych images.

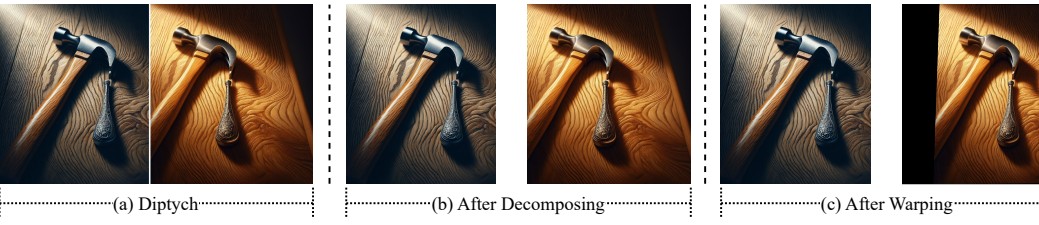

(a) Diptych · · · · · · · · · (b) After Decomposing · · · · · · · · · (c) After Warping

Figure 3: The effect of decomposing and warping in image post-processing.

## 3.3 POST-PROCESSING

After generating the diptych and its corresponding prompt, we implement a tailored post-processing stage aimed at decomposing the diptych back into paired images and further refining the quality of both image pairs and text instructions. This process involves two key steps: **image post-processing** and **instruction refinement**.

**Image Post-processing** The goal of image post-processing is to decompose the diptych into paired images as well as to improve their correspondence. We later use correspondence as a quality control to (optionally) filter our training set. It consists of three steps: *Decomposing*, *Warping*, and *Filtering*:

1. *Decomposing* horizontally separates diptychs generated by DALL-E 3 into image pairs using a retrained object detection model. Specifically, we train a YOLOv8 (Reis et al., 2023) on 3,000 diptych images, where human annotators mark bounding boxes for both left and right segments.

2. *Warping* aligns the decomposed paired images based on semantic correspondence between input and output images. We employ DIFT (Tang et al., 2023), an advanced diffusion-based model, to establish pixel-wise semantic correlations between paired images. By leveraging semantic correspondence, we determine the homography, which maps pixels from the input image to corresponding pixels in the output image, facilitating the precise alignment between them. An example of warping in improving alignment between input and output images is illustrated in Figure 3.

3. *Filtering* assesses image distortion post-warping and retains those with minimal distortion for training purposes. When the dimensions of the image before warping are denoted as $\{w_1, w_2, h_1, h_2\}$, and those after warping as $\{w_3, w_4, h_3, h_4\}$, any image undergoing more than a 50% deformation on any single dimension before and after warping, such as $w_1 < 0.5 * w_3$, is filtered out. Note that this step is applied exclusively to the InstructPix2Pix fine-tuning process for selecting high-quality training samples from our HQ-Edit dataset.

**Instruction Refinement** While image post-processing improves alignment between input and output images, further refinement is vital to ensure that editing instructions are well-aligned with image pairs. First, by leveraging GPT-4V, we rewrite edit instructions based on the differences between input and output image details, thereby enhancing the detail of the text descriptions. Rewriting not only helps fix discrepancies in existing descriptions but also includes visual differences between background objects, which are often omitted in the original text descriptions. Additionally, we use GPT-4V to directly generate inverse-edit instructions for transforming output images back to input images. This simple strategy can effectively double the instruction count but at a marginal cost.

Overall, as demonstrated in Figures 4, the application of rewriting and inversion techniques substantially increases both the length and diversity of edit instructions. This enrichment leads to a dataset enhanced with a wider range of composite operations, resulting in a broader distribution of instruction lengths. Our edit instructions not only have a larger average length but also display a more expansive distribution, underscoring the effectiveness of these augmentation strategies.

## 3.4 DATA QUALITY ASSESSMENT

**Diversity of Edit Instruction** Unlike previous studies which either focus on global or object editing (Brooks et al., 2023; Zhang et al., 2023; 2024), our editing operations span a broad spectrum, encompassing both global operations—such as altering the weather, modifying the background, and transforming the style—and local operations, which include a variety of object-based editing. Figure 5 provides a comprehensive overview of the keywords in the edit instructions of HQ-Edit. This diversity of edit instructions indicates that our HQ-Edit incorporates a vast range of editing tasks, thereby demonstrating its extensive coverage of potential editing operations.

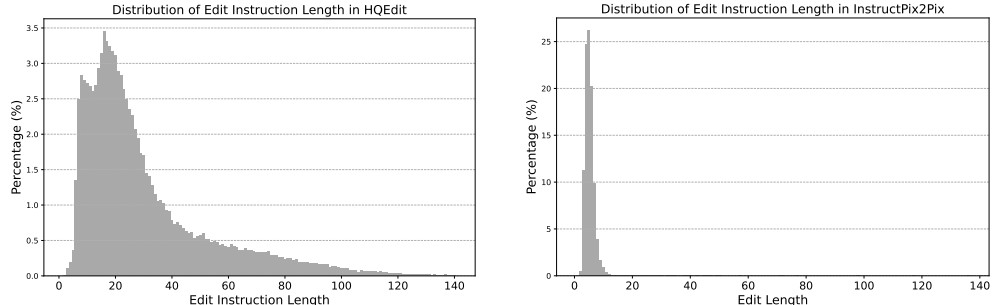

Figure 4: The histograms illustrate the distribution of edit instruction lengths for HQEdit and InstructPix2Pix. Overall, HQEdit exhibits a more uniform and dispersed distribution, indicating a broader diversity in the length of its instructions.

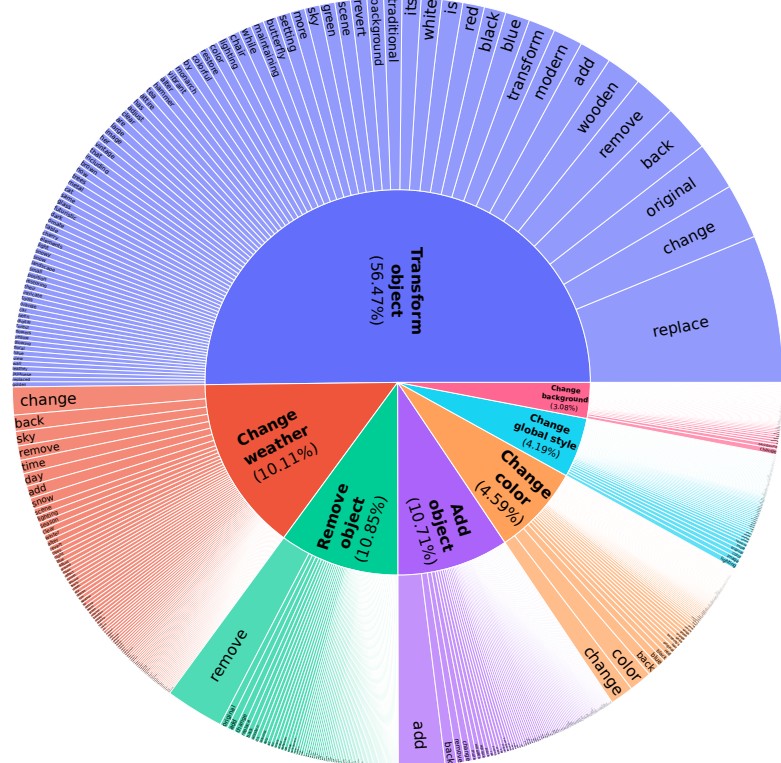

Figure 5: Distribution of edit types and keywords in instructions. The inner ring depicts the types of edit instructions and the outer circle shows the frequency of the instruction keywords.

**Alignment and Coherence** To quantitatively evaluate the quality of editing, we introduce two formal metrics: *Alignment* and *Coherence*. The Alignment metric assesses the semantic consistency of edits with the given prompt, utilizing different criteria for various types of edits, such as global editing (*e.g.*, stylization) and local editing (*e.g.*, object removal), ensuring accurate modifications while preserving fidelity in the rest of the image. On the other hand, the Coherence metric evaluates the overall aesthetic quality of the edited image, considering factors such as lighting and shadow consistency, style coherence, and edge smoothness. These metrics, performed using GPT-4V, produce scores from 0 to 100, with higher scores indicating better alignment or coherence.

We present evaluation example results with varying Alignment scores in Figure 6, and example images showing different Coherence scores in Figure 7, both suggesting a potential (positive) correlation with human perception. Detail of the evaluation can be found at supplemental material.

To further validate the effectiveness of our proposed metrics, as detailed in Section 4.1, we conducted a human evaluation on 1,651 image pairs generated by DALL-E 3. Notably, our metric exhibited a much higher correlation to human preference compared to the popular CLIP score.

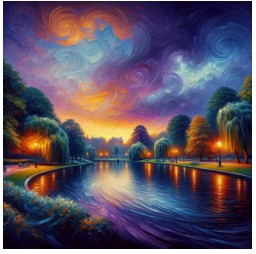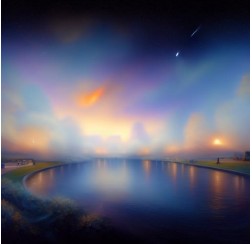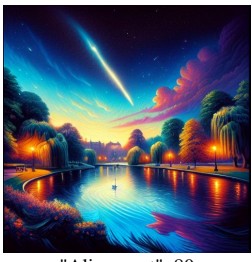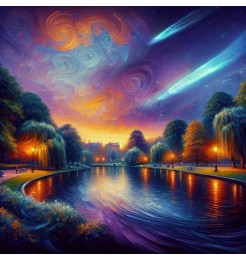

| Input Image | "Alignment": 40 | "Alignment": 80 | "Alignment": 100 |

Figure 6: Examples of different Alignment. Instruction:"*Add a comet in the sky.*" Editing follows the edit instruction more accurately as Alignment increases.

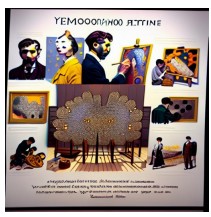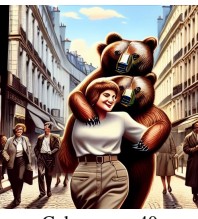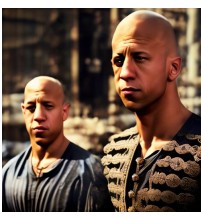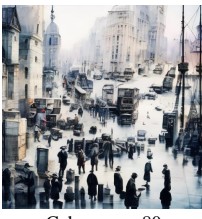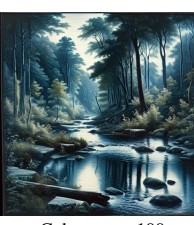

| Coherence : 20 | Coherence : 40 | Coherence : 60 | Coherence : 80 | Coherence : 100 |

Figure 7: Examples of different Coherence. As the Coherence score increases, the image quality improves significantly.

**Comparisons** To demonstrate the superior data quality of HQ-Edit compared to existing public editing datasets, we conduct evaluations on 500 randomly sampled data points from InstructPix2Pix, HIVE, MagicBrush, and HQ-Edit (Table 2), assessing their Alignment and Coherence metrics. HQ-Edit significantly outperforms all others with Alignment and Coherence scores of 92.80 and 91.87, respectively, compared to InstructPix2Pix (68.29 and 83.35), HIVE (9.85 and 84.65), and MagicBrush (80.61 and 65.42), demonstrating its superior data quality.

Table 2: Comparison between different editing datasets.

| Dataset | Alignment ↑ | Coherence ↑ |
|---|---|---|
| InstructPix2Pix (Brooks et al., 2023) | 68.29 | 83.35 |
| HIVE (Zhang et al., 2023) | 9.85 | 84.65 |
| MagicBrush (Zhang et al., 2024) | 80.61 | 65.42 |
| HQ-Edit | 92.80 | 91.87 |

## 4 EXPERIMENTS

**Baselines.** We conducted a comparative analysis with existing open-source text-based image editing methods, *i.e.*, DiffEdit (Couairon et al., 2022), Imagic (Kawar et al., 2023), PromptInverse (Mokady et al., 2022), HIVE (Zhang et al., 2023), MagicBrush (Zhang et al., 2024). To ensure reproducibility and fairness, we utilized default hyperparameters from the official implementations. Our testing set comprised the 293 samples mentioned in Section 3.1, with all input images generated by DALL-E 3 based on the input image descriptions.

**Implementation details.** We choose InstructPix2Pix (Brooks et al., 2023) as our default model, and use HQ-Edit to fine-tune it. During training, we set the image resolution to 512, total training steps to 15000 on 4 A100 GPUs, learning rate to 5e-5, and conditioning dropout prob to 0.05. During the editing, we set the image guidance scale to 1.5, the instruct guidance scale to 7.0, and the number of inference steps to 20.

### 4.1 HUMAN EVALUATION

In this section, we conduct a human evaluation to assess whether GPT-based metrics are good enough and necessary for evaluating the performance of image editing.

**Human Evaluation of Alignment** For alignment, we use CLIP Directional Similarity ($CLIP_{dir}$) (Radford et al., 2021) as the comparison metric, as $CLIP_{dir}$ shares the same purpose as our Alignment metric: verifying the changes in the image are those required by the instruction. The human evaluation is obtained on 1,651 image pairs generated by DALL-E 3. We utilize Gradio (Abid et al., 2019) to create the evaluation platform. For each assessment, edit instructions, the input/output image pairs, and their corresponding descriptions are provided for evaluation. We categorize whether the change

between the input image and the output image matches the corresponding edit instruction into the following 5 levels: **1.** Totally not related; **2.** Not following edit, but there is some relation between the two images; **3.** OK image pair, but not following the edit instruction; **4.** Good image pair, but need to modify the edit instruction for better alignment; **5.** Perfectly follows the edit instruction.

We report the results in Table 3. We use Pearson Correlations to analyze the correlation between Alignment and CLIP$_{dir}$ to Human Evaluation Score. We can observe that the Alignment metric significantly surpasses CLIP$_{dir}$ in accurately evaluating the fidelity of editing instructions to reflect the alterations between the input and output images. This notable discrepancy underscores a significant limitation of CLIP$_{dir}$, namely its inability to comprehensively grasp the nuances of the editing process and accurately retain fidelity to the intricate details of the images.

Table 3: Correlation comparison of Alignment and CLIP$_{dir}$ with Human.

| Alignment | CLIP$_{dir}$ |
|-----------|--------------|
| 0.3592 | -0.1446 |

**Human Evaluation of Coherence** Since traditional metrics for image quality evaluation are lacking, we explored two model-based evaluations: evaluating with 1) general large models like GPT-4 with stronger base capabilities but less specialization or 2) fine-tuned expert models that excel in specific domains despite weaker general capabilities. Specifically, we select ImageReward (Xu et al., 2023), which measures the output image quality, as our expert model comparison metric (Reward Score). As ImageReward's score is unbounded, we normalize it using Sigmoid. We perform human evaluation with 160 image pairs from HQ-Edit — human evaluators are tasked with determining which image in each pair is better. "Image 1 is better", "Tie" and " Image 2 is better" are encoded as -1, 0 and 1. We then compute the Pearson Correlation following the Human Evaluation of Alignment.

Table 4: Correlation comparison of Coherence and Reward Score with Human.

| Coherence | Reward Score | |
|-----------|------|---------|
| | Raw | Sigmoid |
| 0.27 | 0.12 | 0.10 |

As shown in Table 4, the proposed Coherence metric is significantly more aligned with human preference than the Reward Score. This suggests that, as the development of reward models for image generation and image editing is still at a preliminary stage, using a mature and well-tested general-purpose model (like GPT-4V) is more valid and reasonable at this stage.

Table 5: Per model variance of Alignment and Coherence on HQ-Edit test set.

| Alignment | InstructPix2Pix | HIVE | MagicBrush | HQ-Edit |
|-----------|-----------------|------|------------|---------|
| InstructPix2Pix | 0.00 | 12.56 | 14.27 | 12.51 |
| HIVE | 12.56 | 0.00 | 9.87 | 9.21 |
| MagicBrush | 14.27 | 9.87 | 0.00 | 10.51 |
| HQ-Edit | 12.51 | 9.21 | 10.51 | 0.00 |

(a) Alignment variance

| Coherence | InstructPix2Pix | HIVE | MagicBrush | HQ-Edit |
|-----------|-----------------|------|------------|---------|
| InstructPix2Pix | 0.00 | 27.44 | 38.72 | 25.99 |
| HIVE | 27.44 | 0.00 | 32.58 | 26.02 |
| MagicBrush | 38.72 | 32.58 | 0.00 | 36.31 |
| HQ-Edit | 25.99 | 26.02 | 36.31 | 0.00 |

(b) Coherence variance

**Differentiation of GPT-Based Metrics** There still remains a question of whether GPT-4V truly differentiates between scores that are just a few points apart. To this end, we calculate the per-image variance of Alignment and Coherence between InstructPix2Pix, HIVE, MagicBrush, and HQ-Edit, and report the averaged per-image absolute variance in Table 5. We can observe that the per-image variance of Alignment scores are around 10 and Coherence scores around 30, indicating metrics for each image are differentiated by a large margin. This evidence, therefore, can strongly support that our GPT metrics are valid and can accurately tell the quality differences between images.

## 4.2 Quantitative Evaluation on generated images

**Evaluation on HQ-Edit Test Set** We first compare our model with existing text-based image editing models in Table 6 using Alignment and Coherence. Compared to other methods, our model performs best in all metrics. Specifically, our model outperforms the vanilla InstructPix2Pix, achieving a notable increase of 12.30 in Alignment (from 34.71 to 47.01) and 5.56 in Coherence (from 80.52 to 86.16). Furthermore, it is noteworthy that our model surpasses HIVE and MagicBrush, two methods

Table 6: Comparison with existing text-based image editing models on HQ-Edit test set.

| Method | Alignment ↑ | Coherence ↑ |
|--------|-------------|-------------|
| Imagic (Kawar et al., 2023) | 1.50 | 63.58 |
| DiffEdit (Couairon et al., 2022) | 21.53 | 81.81 |
| PromptInverse (Mokady et al., 2022) | 22.82 | 80.85 |
| InstructPix2Pix (Brooks et al., 2023) | | |
| /Base | 34.71 | 80.52 |
| /XL | 35.03 | 84.45 |
| HIVE (Zhang et al., 2023) | | |
| w/conditional | 40.34 | 82.93 |
| w/weighted | 40.68 | 84.94 |
| MagicBrush (Zhang et al., 2024) | 43.77 | 84.19 |
| HQ-Edit | **47.01** | **86.16** |

Table 7: Quantitative Evaluation of Instruct-Pix2Pix Based Method on HQ-Edit Test Set.

| Model | CLIP$_{dir}$ | CLIP$_{img}$ | DINO | SSIM | Reward Score | |
|---|---|---|---|---|---|---|
| | | | | | Raw | Sigmoid |
| InstructPix2Pix | 0.0444 | 0.7353 | 0.7252 | 0.1673 | -0.54 | 0.40 |
| HIVE | 0.0970 | 0.8633 | 0.8851 | 0.4646 | -0.09 | 0.48 |
| MagicBrush | 0.1109 | 0.8173 | 0.8246 | 0.2735 | -0.05 | 0.49 |
| HQ-Edit (Ours) | **0.1351** | **0.9246** | **0.9692** | **0.6561** | **-0.03** | **0.50** |

Table 8: Quantitative Evaluation of Instruct-Pix2Pix Based Method on Emu Edit Test Set.

| Model | CLIP$_{dir}$ | CLIP$_{img}$ | DINO | SSIM | Reward Score | |
|---|---|---|---|---|---|---|
| | | | | | Raw | Sigmoid |
| InstructPix2Pix | 0.0775 | 0.8396 | 0.7879 | 0.2223 | -2.05 | 0.12 |
| HIVE | 0.0527 | 0.8567 | 0.7855 | 0.1978 | -2.07 | 0.12 |
| MagicBrush | 0.1011 | 0.8526 | **0.8278** | **0.2516** | -2.13 | 0.11 |
| HQ-Edit (Ours) | **0.1067** | **0.8588** | 0.8139 | 0.2231 | **-1.97** | **0.13** |

| Input | InstructPix2Pix | HIVE | Magic Brush | HQ-Edit |
|---|---|---|---|---|

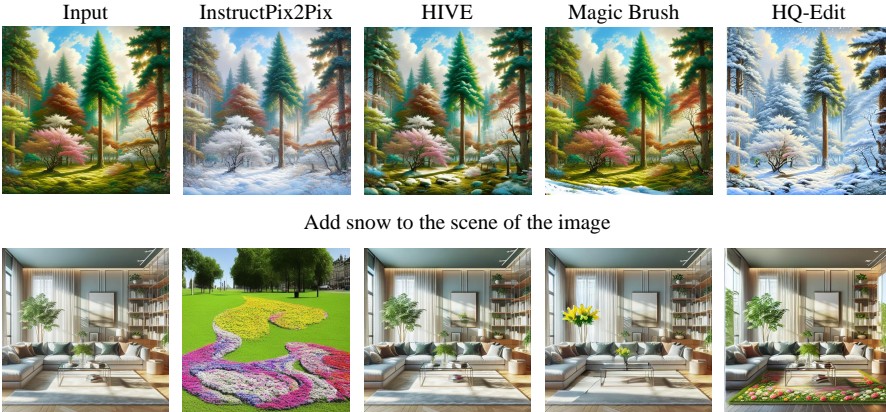

Add snow to the scene of the image

Make flowers on ground region in the image

Figure 8: Qualitative comparison of InstructPix2Pix, MagicBrush, HIVE and HQ-Edit.

| Input | InstructPix2Pix | HIVE | MagicBrush | HQ-Edit |
|---|---|---|---|---|

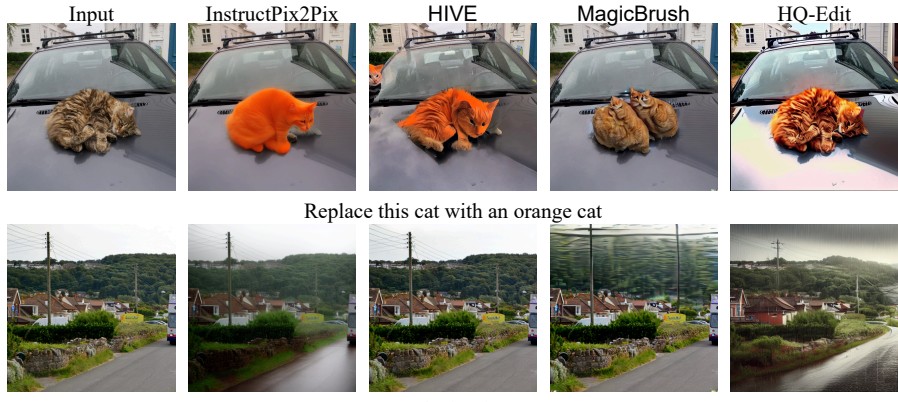

Replace this cat with an orange cat

Make it rainy

Figure 9: Qualitative comparison of InstructPix2Pix, MagicBrush, HIVE and HQ-Edit on Emu Edit

fine-tuned on InstructPix2Pix, further validating its capability to enhance InstructPix2Pix's image editing outcomes beyond their respective datasets. Next in Table 7, we evaluate our model using the most commonly used CLIP$_{dir}$, CLIP image similarity (CLIP$_{img}$), DINO image similarity (DINO), SSIM and Reward Score. HQ-Edit also performs the best in all metrics in the HQ-Edit test set. These results demonstrate our comprehensive corpus of high-fidelity images and precise editing instructions establishes a robust framework for more intuitive and efficacious image manipulation protocols.

**Evaluation on Emu Edit Test Set** We next test on the Emu Edit (Sheynin et al., 2023) which consists of real images. As shown in Table 8, HQ-Edit performs the best in terms of CLIP$_{dir}$, CLIP$_{img}$, and Reward Score and ranks second-best in terms of DINO and SSIM metrics. With these results, we can conclude that the superb quality and large size of HQ-Edit can still enable models to reasonably handle real-life and photorealistic images despite the domain gap.

### 4.3 QUALITATIVE EVALUATION

As shown in Figure 8 and Figure 9, a comparative analysis of the performance of various models is visually presented, with each column dedicated to showcasing the results of a distinct model. In general, we can observe that the models trained with HQ-Edit consistently show stronger visual quality. For example, in the second line of Figure 8, only the model trained with HQ-Edit understands the ground region in the edit instruction and correctly adds the flowers in it as required. It can also be

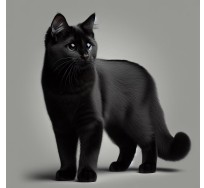 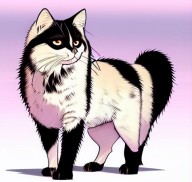 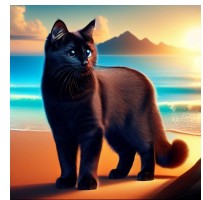 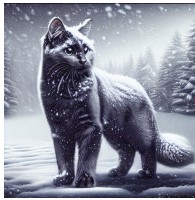 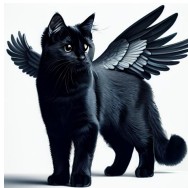

| Input Image | Make the image anime style | Change the background of the subject to beach | Add some snow to the scene of the image | Add wings |
|---|---|---|---|---|

Figure 10: Qualitative results with the same input image but with different edit instructions. HQ-Edit enhances the editing capabilities of InstructPix2Pix by enabling it to modify the same image of a black cat in various ways.

seen in Figure 10 that the model trained with HQ-Edit can carry out various types of edit operations. This observation not only underscores HQ-Edit's advanced understanding of spatial and contextual directives but also its capability to precisely manipulate image content in accordance with specific editing specifications.

## 4.4 ABLATION STUDY

We hereby ablate the effectiveness of different post-processing strategies, introduced in Sec. 3.3. Specifically, we use "RAW" to denote the simply decomposed DALL-E 3 images (*i.e.*, image pairs that directly splitted from diptych), and use "Rewrite", "Filter", "Warp", and "Inverse" to mark whether the corresponding operations are applied for further processing. For example, applying all these four operations to process these will lead to our HQ-Edit dataset. Table 9 reports the corresponding results.

Interestingly, by comparing the first row and the second row, we note that directly fine-tuning the model on the raw DALL-E 3 images enhances its performance on Alignment but hurts Coherence. This potentially suggests that while the image quality of these

Table 9: Ablation experiments on Post-processing.

| RAW | Rewrite | Inverse | Warp | Filter | Alignment ↑ | Coherence ↑ |
|---|---|---|---|---|---|---|
| | | | | | 34.71 | 80.52 |
| ✓ | | | | | 16.83 | 85.74 |
| ✓ | ✓ | | | | 28.62 | 86.68 |
| ✓ | ✓ | ✓ | | | 34.42 | 87.53 |
| ✓ | ✓ | ✓ | ✓ | | 43.41 | 87.56 |
| ✓ | ✓ | ✓ | ✓ | ✓ | 47.01 | 86.16 |

DALL-E 3 generated images exceeds that of the InstructPix2Pix dataset, the alignment between the image and edit instruction is less satisfactory. This issue can be mitigated with our post-processing techniques. For example, our rewrite method, when compared to the second row's results, delivers improvements of 11.79 in Alignment and 0.94 in Coherence. This boost, primarily enhancing the images' alignment with the edit operation, indicates DALL-E 3's challenges in producing accurate images from dypitch prompts—a gap our method effectively bridges. Additionally, employing the inverse technique, which acts as a form of data augmentation, further elevates Alignment by 5.2 and Coherence by 0.94. The warp technique serves to augment both pre- and post-edit image alignment, resulting in a notable 5.2 increase in alignment accuracy. Nonetheless, the application of warp may occasionally lead to undesirable levels of image distortion. Through the implementation of a filtering mechanism targeting such occurrences, we not only achieve a further enhancement in image alignment, registering a 3.6 increase, but also mitigate the associated data volume. Consequently, this filtering process incurs a marginal reduction in Coherence, specifically by 1.4 points, yet remains superior to other baselines. These results indicate that HQ-Edit holds significant potential to enhance instruction-based edit models, especially when combined with effective post-processing.

## 5 CONCLUSION

In this study, we present an automatic way to synthesize the image editing dataset at scale. Specifically, we leverage two foundation models, GPT-4V and DALL-E 3, to automatically generate, rewrite, and expand a set of seed image editing data with *high-quality*. Additionally, we develop two GPT-4V-based evaluation metrics to assess the alignment of the edited images to the editing instruction, and the coherence of the image content. Our extensive experiments demonstrate that models trained on HQ-Edit set a new state-of-the-art performance in the task of instruction image editing.

**Ethical Discussion** The integration of foundation models like GPT-4 and DALL-E into automatic dataset creation offers significant benefits by enhancing data quality, diversity, and the overall coverage of editing operations, which in turn supports rapid, large-scale generation of precise image-edit pairs for training advanced image editing models. However, this same automation also introduces ethical challenges, as AI-generated content may inadvertently embed biases, stereotypes, or other unintended issues due to reduced human oversight and a lack of clear data provenance. Additionally, this uncertainty complicates efforts to verify authenticity, uphold privacy norms, and prevent the spread of misinformation or the reinforcement of societal biases. To address these challenges, future work should prioritize the development of verifiable, transparent data-generation pipelines that integrate systematic bias detection and mitigation strategies. Incorporating human validation where possible can further safeguard against inadvertent harms, enabling regular audits and ensuring that ethical standards are met throughout the data-generation process. We believe strengthening accountability through these measures will enable us to responsibly harness the innovative potential of these foundation models while simultaneously promoting fair, trustworthy, and socially responsible applications of image editing in the real world.

**Acknowledgement** We thank the AWS Cloud Credit for Research Program, the Microsoft Accelerate Foundation Models Research Program, and the OpenAI Researcher Access Program for supporting our computing needs.

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

# A APPENDIX

# B PROMPTS

We list all the prompts we used for data collection, including the EXPAND PROMPT used for the Expansion step; DIPTYCH PROMPT and REWRITE PROMPT used for the Generation step; and two metric prompt ALIGNMENT PROMPT and COHERENCE PROMPT for the evaluation.

## B.1 STEP #1: EXPANSION

---

**EXPAND PROMPT (GPT-4)**

You are required to generate num examples considering the given examples. The examples should vary widely, including different human characteristics (such as race, age, and body type), various animals, insects, furniture, tools, or any object types, etc., and diverse backgrounds (like different countries, natural environments, landscapes, or skies). The editing attributes should also be diverse. Make sure the examples are clear, concise, comprehensive, and easier for DALL-E 3 to generate this diptych image following the prompt. Describe the first image in "INPUT_DESCRIPTION" like "input", the second image in "OUTPUT_DESCRIPTION" like "output", both "INPUT_DESCRIPTION" and "OUTPUT_DESCRIPTION" should be independent complete sentences, and the operation that edits the first image to the second image in "EDIT_OPERATION", and the operation that edits the second image to the first image in "INVERSE_EDIT_OPERATION", the output should be a list of JSON format as such:
{ "input": "INPUT_DESCRIPTION",
"edit": "EDIT_OPERATION",
"edit_inv": "INVERSE_EDIT_OPERATION",
"output": "OUTPUT_DESCRIPTION". }.
Do not output anything else, all examples should have complete keys "input", "edit", "edit_inv", and "output".

---

## B.2 STEP #2: GENERATION

---

**REWRITE PROMPT (GPT-4)**

Please rewrite the following prompt to make it more clear and concise, and easier for DALL-E 3 to generate this diptych image follow the prompt. The original prompt is: {prompt}. The output prompt should start with "REVISED":

---

**DIPTYCH PROMPT (DALL-E 3)**

Create a diptych image that consists two images. The left image is {prompt}; The right image keep everything the same but {edit_action}.

---

## B.3 EVALUATION METRIC

---

**ALIGNMENT PROMPT (GPT-4V)**

From 0 to 100, how much do you rate for EDIT TEXT in terms of the correct and comprehensive description of the change from the first given image to the second given image? Correctness refers to whether the text mentions any change that are not made between two images. Comprehensiveness refers to whether the text misses any change that are made between two images.

---

The second image should have minimum change to reflect the changes made with EDIT TEXT. Be strict about the changes made between two images:
1. If the EDIT TEXT is about stylization or lighting change, then no content should be changed and all the details should be preserved.
2. If the EDIT TEXT is about a local change, then no irrelevant area nor image style should be changed.
3. The first image should not have the attribute described inside the EDIT TEXT, rate low, (<80) if this happens.
4. Be aware to check whether the second image does maintain the important attribute in the left image that is not reflected in the EDIT TEXT. Rate low (<50) if two images are not related.
Provide a few lines for explanation and give the final response in a json format as such:
{ "Explanation": "",
"Score": "", }

---

### COHERENCE PROMPT (GPT-4V)

Rate the Coherence of the provided image on a scale from 0 to 100, with 0 indicating extreme disharmony characterized by numerous conflicting or clashing elements, and 100 indicating perfect harmony with all components blending effortlessly. Your evaluation should rigorously consider the following criteria:
1. Consistency in lighting and shadows: Confirm that the light source and corresponding shadows are coherent across various elements, with no discrepancies in direction or intensity.
2. Element cohesion: Every item in the image should logically fit within the scene's context, without any appearing misplaced or extraneous.
3. Integration and edge smoothness: Objects or subjects should integrate seamlessly into their surroundings, with edges that do not appear artificially inserted or poorly blended.
4. Aesthetic uniformity and visual flow: The image should not only be aesthetically pleasing but also facilitate a natural visual journey, without abrupt interruptions caused by disharmonious elements.

Implement a stringent scoring guideline:
- Award a high score (90-100) solely if the image could pass as a flawlessly captured scene, devoid of any discernible disharmony.
- Assign a moderate to high score (70-89) if minor elements of disharmony are present but they do not significantly detract from the overall harmony.
- Give a moderate score (50-69) if noticeable disharmonious elements are evident, affecting the image's harmony to a moderate degree.
- Allocate a low score (30-49) for images where disharmonious elements are prominent, greatly disturbing the visual harmony.
- Reserve the lowest scores (0-29) for images with severe disharmony, where the elements are so discordant that it disrupts the intended aesthetic.

Your assessment must be detailed, highlighting the specific reasons for the assigned score based on the above criteria. Conclude with a response formatted in JSON as shown below:
{ "Explanation": "<Insert detailed explanation here>",
"Score": <Insert precise score here> }

## C  MORE VISUALIZATION RESULTS

### C.1  DATA POINTS

Here, we provide two randomly sampled data points from HQ-Edit in Figure 11 for visual assessment

**"input":** "A German Shepherd with a black and tan coat, pointed ears, and a dog tag is sitting on a grassy lawn with trees and sunlight in the background. "

**"input":** "portrait of a majestic golden eagle in flight against a warm sunset sky."

**"edit":** "Replace the German Shepherd with a fluffy white cat with blue eyes and a long tail, keeping the same background. "

**"edit":** "adjust the overall image to a nighttime setting with a darker sky and cooler tones."

**"inverse-edit":** "Replace the fluffy white cat with the original German Shepherd with a black and tan coat, pointed ears, and a dog tag. "

**"inverse-edit":** "adjust the overall image to a daytime setting with a brighter sky and warmer tones."

**"output":** "A fluffy white cat with blue eyes and a long tail is sitting on the same grassy lawn with trees and sunlight in the background."

**"output":** "portrait of a majestic golden eagle in flight against a cooler, night-time sky with visible clouds and a darker ambiance."

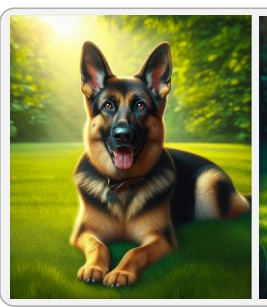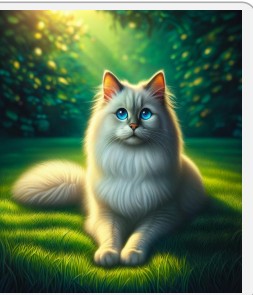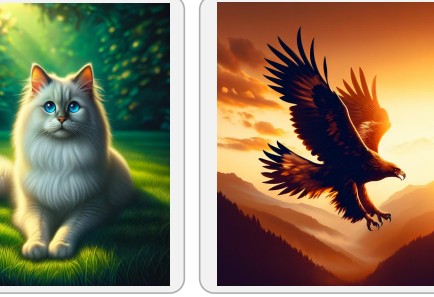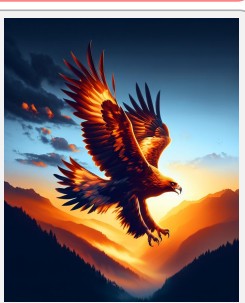

"Alignment": 100 "Coherence":100          "Alignment": 60 "Coherence":100

Figure 11: Example data sampled from HQ-Edit. Our data contains two main parts, Instruction (input, edit, inverse-edit, output) and Image (input image, output image). The two samples highlight that, 1) the image is densely packed with details, 2) the input and ouput offers a comprehensive description of the input and output image, and 3) the edit and inverse-edit instructions precisely delineate the transformations occurring between the two images.

## C.2    DATA POINTS COMPARISON

We visualize the data of InstructPix2Pix in Fig. 13, of MagicBrush in Fig. 14, of HIVE in Fig. 15, and HQ-Edit in Fig. 12 with the Edit instruction, Aligment and Coherence. This shows that HQ-Edit possesses higher image quality and better image-text alignment.
and more data with its Alignment and Coherence score from HQ-Edit in Figure 12.

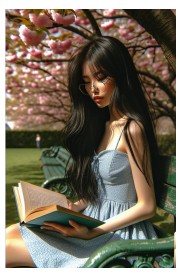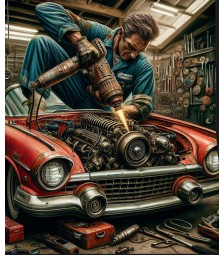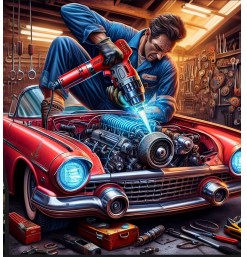

Edit: change her hair color to blonde and add waves to it
Alignment: 100
Coherence: 95

Edit: Replace the heavy-duty power drill with a high-tech precision power tool.
Alignment: 100
Coherence: 95

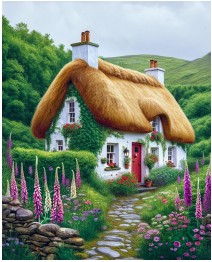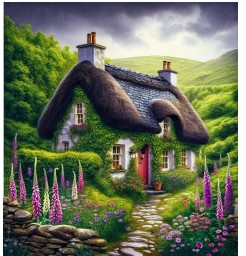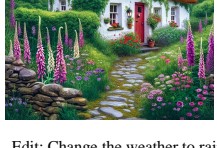

Edit: Change the weather to rainy.
Alignment: 100
Coherence: 95

Edit: Transform the elderly woman into a young woman, change her traditional dress to a modern black leather jacket, replace her sandals with white sneakers, and add a black handbag beside her on the bench.
Alignment: 100
Coherence: 90

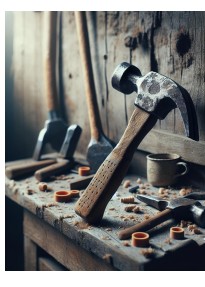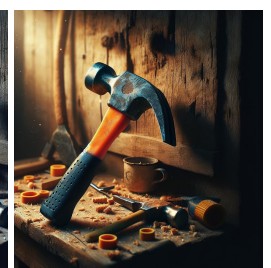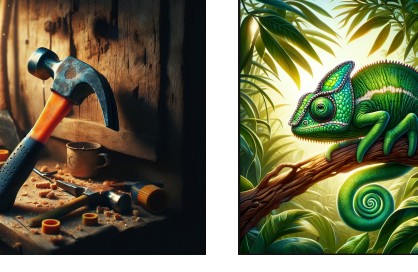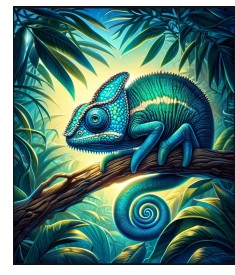

Edit: Replace the metal hammer with a plastic toy hammer with a bright orange and blue handle.
Alignment: 80
Coherence: 95

Edit: Change the chameleon's body to a vivid blue hue while keeping the green color on its head crest and tail.
Alignment: 100
Coherence: 100

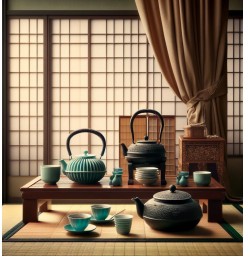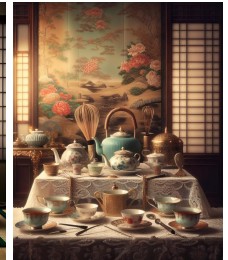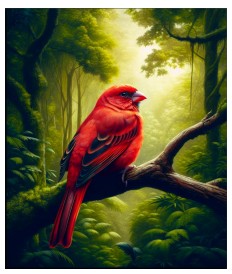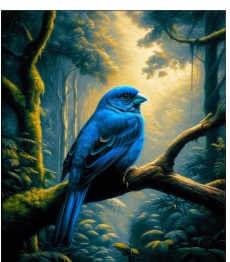

Edit: Replace the Japanese tea set with a Victorian tea set, including porcelain teapots and cups with floral designs, add a lace tablecloth, silver cutlery, and a decorative golden tea strainer. Change the backdrop to include a framed floral tapestry.
Alignment: 100
Coherence: 88

Edit: Alter the bird's color to vibrant blue.
Change the backdrop to include a framed floral tapestry.
Alignment: 100
Coherence: 95

Figure 12: Data of HQ-Edit, the left side is the input image and the right side is the output image.

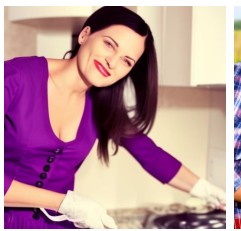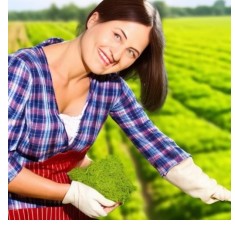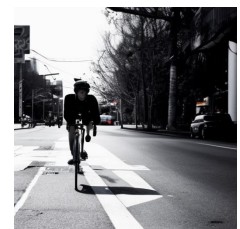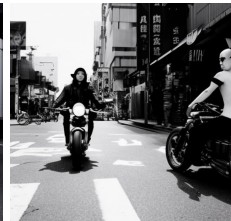

Edit: Make her a farmer
Alignment：80
Coherence：65

Edit: swap the cyclist for a biker
Alignment：40
Coherence：90

Figure 13: Data of InstructPix2Pix, the left side is the input image and the right side is the output image.

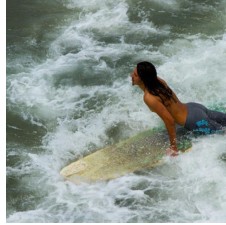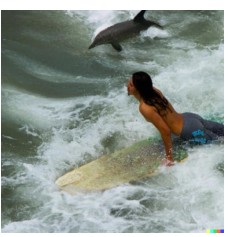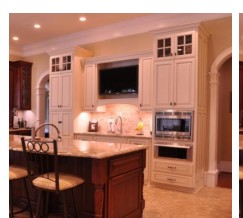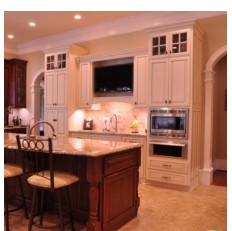

Edit: Add a dolphin jumping out of the water
Alignment：100
Coherence：75

Edit: Turn on the faucet
Alignment：0
Coherence：95

Figure 14: Data of MagicBrush, the left side is the input image and the right side is the output image.

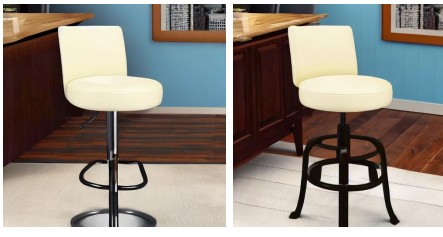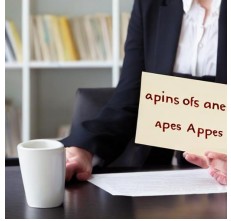

Edit: Change retro to futuristic
Alignment：85
Coherence：95

Edit: make the man a woman
Alignment：50
Coherence：30

Figure 15: Data of HIVE, the left side is the input image and the right side is the output image.

