# OpenReview forum: "HQ-Edit: A High-Quality Dataset for Instruction-based Image Editing"
_ICLR.cc/2025/Conference — ICLR 2025 Poster_

### Official Review · Reviewer_x6Ut · 2024-11-02

**Soundness:** 2
**Presentation:** 2
**Contribution:** 2
**Rating:** 3
**Confidence:** 5

**Summary:**

This paper presents a new high resolution dataset on image editing named HQ-Edit. Compared to existing works, the proposed dataset contains high resolution and leveraged the most advanced vision-language model for creating human like instructions. Since all the instruction and image generated are using synthetic data, as a result, the proposed dataset is in a reasonable large scale for public usage. Simply leverage the dataset, the existing InstructPix2Pix can generate better results.

**Strengths:**

The key contribution in this paper is the proposed HQ-Edit dataset. Compared to existing widely used InstructPix2Pix dataset, HQ-Edit dataset is in a high resolution and better editing due to verification  process of a powerful vision-language models such as GPTV.  For example, after generating the dataset by Dalle 3.  GP4V will be used to rewrite the instructions to improve the instruction and editing alignment. Such process will greatly improve the dataset.  Besides, human evaluation has shown that the proposed dataset is better on editing image quality.

**Weaknesses:**

The key weakness of the proposed method lies on the limited quality improvement of dataset.

First, InstructPix2Pix firstly proposed a pipeline to leverage prompt2prompt and gpt to generate the triplets. While the proposed "high quality" editing dataset is simply replaced the prompt2prompt method to Dalle3 and leveraged a more powerful gpt version to filter out the low quality instructions. Such improvement is incremental while not significantly improve the datasets.  The main reason is, similar to prompt2prompt, the HQ Edit dataset is still all synthetic pairs.  The proposed editing types are still similar to InstructPix2Pix.  Most importantly, when most of existing editing dataset such as Magicbrush, SeedX and UltraEdit have adopt real image as input,  the usage of synthetic image pairs has became less important.

Second, the proposed dataset have shown multiple editing type, while leveraging Dalle3 is obviously not the optimal solutions.  SeedX has been demonstrated the remove and add case should leverage more accurate model such as image inpainting. Meanwhile, the quality of Dalle3 images have become less convincing to be recognize as "high resolution and high quality" since Flux and other generative super resolution can achieve much better higher resolution image editing.

**Questions:**

Since the proposed dataset contains multiple type of editing, it is important to show the results of different editing cases, especially on real images.   Moreover, it is also better to compare with some specialist models such as inpainting and object insertion model.

In addition, I also notice that the edited results still can't keep the un-edited region unchanged. Many cases showing in the paper indicate that changing unrelated region will cause lots of identity shift on other regions.

It also worths to show more global editing on more real images such as stylization (painting, sketch, water color, cartoon style) and image2image translation (day to night, winter to summer). EMU edit benchmark should be a good dataset to be evaluated.

Reference
[1]https://huggingface.co/datasets/AILab-CVC/SEED-Data-Edit
[2]https://huggingface.co/datasets/BleachNick/UltraEdit

**Details Of Ethics Concerns:**

The proposed method can edit human skin color and face expressions.  It is worth to discuss it but those discussion is missing in the paper.

---

> ### Author Response · Authors · 2024-11-25
>
> We thank the reviewer for the detailed comments and find that our HQ-Edit is of a high resolution and better editing compared to the existing widely used InstructPix2Pix, which is supported by human evaluation. We address your concerns as follows:
>
> ## Incremental improvements compared to InstructPix2Pix
> We apologize for any confusion regarding our contribution. As explained in the **Common Concerns/Necessity of diptych generation**, DALL-E 3 and Prompt2Prompt differs significantly in functionality as:
>     1. DALL-3 API can not be naively utilized to do image-to-image generation.
>     2. In InstructPix2Pix, image consistency is achieved via manipulating the attention matrix of the generation model, which is impossible with any close-sourced model. In contrast to InstructPix2Pix, the pipeline of HQ-Edit is universal and can utilize any image generation model.
>
> These restrictions motivate us to take diptych generation as a workaround and develop a curated post-processing pipeline so that such a powerful model can be utilized effectively and at scale.
>
> Please see **Common Concerns/Necessity of diptych generation** for more.
>
> ## Usage of synthetic image pairs has become less important
>
> Thank you for raising this concern. We would like to highlight that an important contribution of our paper is providing a high-quality and scalable pipeline to collect data from powerful (closed-source) text-to-image (T2I) models. While our specific target is DALL-E 3 in this paper, we clarify that our pipeline can be easily adapted to other T2I models, such as the recent FLUX or Midjourney, which can generate more photorealistic images.
>
> Next, following your suggestion, we evaluated our approach on the EMU Edit benchmark to demonstrate that, even with DALL-E 3, our results remain competitive:
>
> | Model          | $\text{CLIP}_\text{dir}$ | $\text{CLIP}_\text{img}$ | DINO        | SSIM           | Reward Score | Reward Score (after Sigmoid) |
> |---------------|---------------------------|--------------------------|----------------|----------------|--------------|-----------------------------|
> | InstructPix2Pix    | 0.0775                    | 0.8396                  | 0.7879         | 0.2223         | -2.05        | 0.12                  |
> | HIVE               | 0.0527                    | 0.8567                  | 0.7855         | 0.1978         | -2.07        | 0.12                  |
> | MagicBrush         | 0.1011                    | 0.8526                  | **0.8278**     | **0.2516**     | -2.13        | 0.11                  |
> | HQ-Edit (Ours)     | **0.1067**                | **0.8588**              | 0.8139         | 0.2231         | **-1.97**    | **0.13**              |
>
> As shown in the table, HQ-Edit performs the best in terms of CLIP_dir, CLIP_img, and Reward Score (Jiazheng et al. [1]) and ranks second-best in terms of DINO and SSIM metrics. Some qualitative results are shown in the `real_image_results.png` in this [anonymous repo](https://anonymous.4open.science/r/hqedit_re-8D01). With these results, We conclude that the superb quality and large size of HQ-Edit can still enable models to handle real-life and photorealistic images reasonably well.
>
> Lastly, compared to collecting real images, we highlight that using synthetic images can be more scalable and comprehensive. We believe that a stronger editing dataset could be a mixture of synthetic and real images, where synthetic images contribute to comprehensiveness and precision—properties that may be challenging to achieve with real images alone.
>
> We will add these discussions and the results in the revision.
>
> Please see **Common Concerns/Performance on real images (EMU Edit)** for more.
>
> ## Expert models produce better training data
> Thanks for presenting this idea. Using expert models such as image inpainting models can yield better results in certain areas. However, this would significantly limit the range of editing operations. As illustrated in Fig. 1 and Fig. 11, our HQ-Edit process requires a lot of long and complicated editing instructions, which are far beyond the scope of expert models that are designed for very certain kinds of tasks. These complicated editing instructions in the training set also contribute to our finetuned models' performance.
>
> ## Results showing un-edited regions are changed
> We appreciate you bringing up this concern. We acknowledge that altering unedited regions is a common and open issue in instruction-based image editing, regardless of the datasets used for training. We hope this issue can be mitigated in the future through improved instruction-editing frameworks, such as enhanced prompt design and architectural changes.

---

> > ### Author Response · Authors · 2024-11-25
> >
> > ## More global editing on more real images such as stylization and image2image translation
> > Thanks for your suggestion. We present some qualitative results of stylization (Sketch, Cartoon) and image2image translation (Rain, Night) in `real_image_results.png` in this [anonymous repo](https://anonymous.4open.science/r/hqedit_re-8D01). One can notice that the model finetuned with HQ-Edit produces significantly better results compared to models trained with InstructPix2Pix, MagicBrush, and HIVE.
> >
> > ## Ethic concern
> > Since our dataset is generated using prompts from DALL-E 3, which has a stringent built-in safety protocol, we are confident that HQ-Edit contains only ethical data. We appreciate the emphasis on ethical considerations and will ensure that discussions around editing human skin color and facial expressions are included in the revision.
> >
> > [1] Xu, J., et al. (2023). ImageReward: Learning and Evaluating Human Preferences for Text-to-Image Generation. In NeurIPS 2023

---

> ### Comment · Reviewer_x6Ut · 2024-11-26
> **This paper has a fundamental problem on image editing dataset generation.  Dall-E 3 is not suitable for generating image editing pairs with identity preservation.**
>
> Thanks for providing new evidences. Although some of my opinions have been changed, the major concerns are still not be addressed.
>
> 1.  Given the Dall-E 3 generation pipeline, we need discuss whether it is the optimal way for generated high quality image editing dataset ? From current rebuttals, it is clear that the answer is no.  Since there is no guarantee the data will contain high quality image pairs, therefore, no matter how well the filtering function it is, there is no way to result a high quality dataset.  As I mentioned,  I have seen lots of background and identity shift from the examples. Especially the newly provided examples give lots evidences to support my opinion.  The reason behind that is the dataset itself has some bias for identity preservation.  For example, replacement, view point change. even remove, all the results suffer id lost.  Thus it is more like a image generation rather than image editing.   The Dall-E3  is a text 2image model, it dose not learn how to preserve the image content.  This is the main reason.
>
> 2. How dose the proposed method dataset applied to real world image editing?  It is still questionable. There is no strong evidence on it.    Given the newly showing results, I can observe that most examples are not useful in real world application.  For example,  1 the hood of the car is not visually plausible.  2. make it rainy but changes whole thing. 3  changed the dog is still a dog. 4 Hydrant background totally changed. 5 removed refrigerator but adding a new one. 6,7,8 the image is not a sketch ....
> Only last three results look OK.
>
> I am open to change my mind but I could not find many promising results to support it.

---

> ### Author Response · Authors · 2024-11-27
>
> Thanks for your responses. However, we respectfully disagree with your opinions and would like to emphasize why we believe this work is valuable:
>
> ----
> Firstly, as generative models become increasingly powerful, many of them remain closed-source. We believe it is both interesting and important to leverage their APIs to advance research within the open-source community. Our HQ-Edit is an example of this approach, demonstrating how to create a pipeline for generating high-quality datasets for instruction-based image editing. Our extensive experiments confirm its effectiveness: for instance, using our own dataset, we observe consistent and substantial improvements over other methods; with EMU Edit, we achieve competitive performance on real images. This suggests that well-designed synthetic data generation pipelines can be a viable alternative to human-annotated datasets in advancing image editing capabilities.
>
> Moreover, we would like to clarify that our work can be easily extended to models other than DALL-E 3. For example, we believe that by applying our framework to more recent text-to-image models like FLUX, which can generate highly photorealistic images, we can produce an even stronger dataset to support instruction-based editing.
>
> With these points in mind, we believe our work is valuable, and our experiments using DALL-E 3 serve as a valid proof-of-concept supporting the effectiveness of our developed framework.
>
> Next, we would like to point out that your judgment of our generated image quality may not fully consider the comparative context. The appropriate way to assess our generated images is by evaluating them among all candidate models. To this end, we present per-image metrics (i.e., Alignment, Coherence, and CLIP_dir) below. We observe that our approach consistently maintains the highest win rate across all metrics, suggesting the effectiveness of our model. To create images that better satisfy visual judgments, leveraging larger models, longer training times, and other enhancements could help. However, this is beyond the focus of this paper.
>
>
> | Alignment | Avg. | Win Rate | Image 1 | Image 2 | Image 3 | Image 4 | Image 5 | Image 6 | Image 7 | Image 8 | Image 9 | Image 10 | Image 11 | Image 12 |
> | - | - | - | - | - | - | - | - | - | - | - | - | - | - | - |
> | InstructPix2Pix | 70.83 | 41 | 30 | 90 | 70 | 10 | 65 | **95** | 20 | **90** | 85 | **100** | **100** | **95** |
> | HIVE | 41.25 | 16 | 40 | 40 | 20 | 20 | **90** | **95** | 20 | 40 | 60 | 40 | 20 | 10 |
> | MagicBrush | 47.5 | 16 | 40 | **95** | 60 | 90 | **90** | 20 | 10 | 20 | 20 | 10 | 60 | 55 |
> | HQ-Edit (Ours) | **79.58** | **66** | **95** | 20 | **90** | **100** | 30 | **95** | **60** | **90** | **90** | 95 | 95 | **95** |
>
> | Coherence | Avg. | Win Rate | Image 1 | Image 2 | Image 3 | Image 4 | Image 5 | Image 6 | Image 7 | Image 8 | Image 9 | Image 10 | Image 11 | Image 12 |
> | - | - | - | - | - | - | - | - | - | - | - | - | - | - | - |
> | InstructPix2Pix | 68.67 | 23 | 62 | 81 | 64 | **88** | 46 | 81 | 74 | 78 | **48** | 61 | 71 | **70** |
> | HIVE | 63.83 | 7 | 55 | 75 | 35 | 76 | 35 | **85** | 85 | 85 | 45 | 85 | 75 | 30 |
> | MagicBrush | 61.67 | 23 | 35 | 35 | **92** | 45 | 52 | 81 | **90** | 85 | 45 | **95** | 45 | 40 |
> | HQ-Edit (Ours) | **72.67** | **46** | **85** | **85** | 65 | 55 | **85** | 65 | 85 | **90** | 35 | 75 | **82** | 65 |
>
> |  $\text{CLIP}_\text{dir}$ | Avg. | Win Rate | Image 1 | Image 2 | Image 3 | Image 4 | Image 5 | Image 6 | Image 7 | Image 8 | Image 9 | Image 10 | Image 11 | Image 12 |
> | - | - | - | - | - | - | - | - | - | - | - | - | - | - | - |
> | InstructPix2Pix | 0.1095 | 17 | 0.1847 | 0.1170 | 0.1249 | 0.1269 | 0.0259 | 0.0993 | **0.0851** | **0.1721** | 0.1216 | 0.0385 | 0.0482 | 0.1703 |
> | HIVE | 0.1094 | 8  | 0.1700 | 0.0142 | 0.2868 | 0.1845 | 0.0091 | **0.1957** | 0.0119 | 0.1434 | 0.0922 | 0.0162 | 0.0590 | 0.1294 |
> | MagicBrush | 0.1006 | 17 | 0.1153 | 0.0171 | **0.3166** | 0.0601 | 0.0418 | 0.0528 | 0.0003 | 0.1620 | 0.0320 | **0.0295** | 0.2310 | 0.1981 |
> | HQ-Edit (Ours)  | **0.1473** | **58** | **0.2224** | **0.1378** | 0.1602 | **0.2362** | **0.0909** | 0.0468 | 0.0811 | 0.1234 | **0.1291** | 0.0291 | **0.2332** | **0.2768** |
>
> We are happy to answer if you have any other questions.

---

> ### Comment · Reviewer_x6Ut · 2024-11-28
>
> As a researcher in image editing for many years. I want to express my deeply concerns again. I have emphasized many times in the discussion, Image editing should consider  $\textbf{identity preservation}$.  This is KEY difference to image generation.  No matter which method, Dall-E3 or Flux. is used, if we only leverage prompts to generate a pair of editing images, it will be extremely difficult.  HQ-Edit has demonstrated that with carefully design of filtering and prompting, the trained model has little value to improve the editing performance as most of real world image editing results are not acceptable.  Please consider the human perceptual results rather than the metrics.   All of these metrics can not fully indicate the results. Please explain my concerns on the newly presented failures.  Please explain why the HQ-Edit claimed many editing types but replacement, inpaiting, insertion,  and camera move/view point change can't give stratifying results.  I think the dataset quality is not good enough otherwise those cases should not be failed.   Prompt 2 prompt can handle  both local editing and global editing,  even though the quality  is not good enough, it still has lots of identity well preserved cases with good filtering metrics.  In terms of HQ-Edit, we can clearly see the identity preservation is  main problem, it can't be addressed as the data generation pipeline has a fundamental problem: Dalle3 can't generate identity preserved editing pairs on object replacement,  inpatinting, object insertion and etc,.  Moreover, through API, the results has very limited control and this is reason that I don't believe the dataset will be useful for future research. Leveraging prompt control for image generation methods to generated a high quality local editing pairs is very difficult.   HQ-Edit dataset claimed editing coverage is way over it can actually handle.

---

> > ### Comment · Reviewer_PppU · 2024-11-28
> > **Official Comment by Reviewer PppU**
> >
> > I completely agree with the reviewer's opinion. After reviewing the three additional results provided by the authors, I noticed significant changes in the background area of the edited images. I have lowered my rating, but I am more than willing to raise it if the authors can provide an effective solution.

---

> > > ### Author Response · Authors · 2024-12-01
> > >
> > > Dear Reviewer PppU and x6Ut
> > >
> > > We provide a general response to these questions here https://openreview.net/forum?id=mZptYYttFj&noteId=UVE8tLBjAJ
> > >
> > > Please let us know if it addresses your concerns.
> > >
> > > Thanks
> > > Authors

---

> > > > ### Comment · Reviewer_x6Ut · 2024-12-02
> > > >
> > > > Thanks for sharing the new results.  I will give a clear accept for HQ-Edit if it can have similar results to commercial model.  However, the gap is huge and I want to make it transparent to the research community, could you share some details of the new model without disclosuring the secrets?  For example, to rate the current paper, I want to know how much HQ-Edit data have been used for training, how much additional data have been used. Whether those new data are using Dall.E3 for generation?  I am concerning that the new model leverages more data which is not described in the current draft. I agree that filter function and metric are useful but the HQ-Edit dataset maybe not.  I also feel that the current details described in the draft may not be enough to cover the new model capability.  Only leverage the data pipeline and existing data with scaling up model architecture  won't achieve such good results.  It will be tricky if I rate the paper based on this new model results.  We will discuss with it with AC to figure it out if those information are not available.

---

> ### Author Response · Authors · 2024-12-03
> **HQEdit helps motivate the commercial model.**
>
> Dear Reviewer x6Ut,
> We may not disclose the exact data strategy. However, based on this limited tech report that we can share [https://anonymous.4open.science/r/hqedit_re-8D01/SeedEdit_Align_Image_Re_Generation_to_Image_Editing.pdf ], we assert that a significant portion of the data used in the SeedEdit model is regenerated or sampled using an internal commercialized T2I model, following similar principles to [diptych generation]. This approach is also independently acknowledged in recent studies, such as those proposing in-context LoRA [ https://arxiv.org/pdf/2410.23775 ] and OminiControl [ https://arxiv.org/pdf/2411.15098. Check A.1. Generation pipeline]
>
> Overall, we could foresee that HQEdit has the potential to serve as a foundational strategy and provide strong motivation for diverse data generation. From many perspectives, transferring the methodology from HQEdit to more advanced released models, such as FLUX, can significantly enhance data performance and narrow the performance gap for real image applications.

---

### Official Review · Reviewer_Drub · 2024-11-03

**Soundness:** 3
**Presentation:** 3
**Contribution:** 3
**Rating:** 5
**Confidence:** 3

**Summary:**

This paper presents HQ-Edit, a dataset designed for instruction-based image editing, which contains around 200,000 high-quality image edits. The authors have developed a scalable data collection pipeline that utilizes advanced foundation models, GPT-4V and DALL-E 3, to generate detailed text prompts paired with input and output images. This approach overcomes the limitations of previous methods that relied on attribute guidance or human feedback, which were constrained by scalability and diversity.

To quantitatively assess the quality of image-edit pairs, the authors introduce two novel metrics: Alignment and Coherence. Alignment evaluates the semantic consistency with edit prompts, while Coherence assesses the overall aesthetic quality of the edited image.

The author fine-tuned the InstructPix2Pix model using the HQ-EDIT dataset.Compared to existing text-based image editing models,their model performs best in Alignment and Coherence Score.

**Strengths:**

1. High-Quality Dataset: The paper introduces HQ-Edit, a dataset with approximately 200,000 high-quality image edits, which is a significant contribution to the field of instruction-based image editing.
2. Advanced Foundation Models: Leveraging state-of-the-art models like GPT-4V and DALL-E 3 ensures that the dataset benefits from the latest advancements in AI, leading to high-resolution and detailed images.
3. Broad Coverage of Editing Operations: HQ-Edit covers a wide range of editing tasks, from global operations like weather changes to local object edits, demonstrating its versatility.

**Weaknesses:**

1. Synthetic Data Limitations: Although the synthetic images are useful for training, the trained model may not perform well on real images.
2. Constrained Persuasiveness of Evaluation Metrics:The paper only conducted comparisons on the two evaluation metrics it proposed, Alignment and Coherence, without making comparisons on more widely used and popular metrics.

**Questions:**

1. In the appendix, the provided image scores suggest that GPT-4V seems to give higher coherence scores to synthetic images. The paper does not substantiate that higher coherence scores from GPT-4V are indicative of higher human ratings.This raises the question of whether GPT-4V is 、capable of assessing image quality accurately.
2. Both evaluation metrics have a scale ranging from 0 to 100. Even though there are some segmented scoring prompts, can GPT-4V truly differentiate between scores that are a few points apart?

---

> ### Author Response · Authors · 2024-11-25
>
> We thank the reviewer for the detailed comments and find that our HQ-Edit is of high quality and constructed with very advanced foundation models, e.g., GPT-4V and DALL-E 3, and covers a variety of editing operations. We address the concerns as follows:
>
> ## Gap between real-life images and synthetic images
> Thank you for raising this concern. Yes, this is one limitation of our work at this stage. To better understand how our model handles real images, we first evaluate on the test set of EMU Edit, which consists only of real-life images.
>
> | Model          | $\text{CLIP}_\text{dir}$ | $\text{CLIP}_\text{img}$ | DINO        | SSIM           | ImageReward Score | ImageReward Score (after Sigmoid) |
> |---------------|---------------------------|--------------------------|----------------|----------------|--------------|-----------------------------|
> | InstructPix2Pix    | 0.0775                    | 0.8396                  | 0.7879         | 0.2223         | -2.05        | 0.12                  |
> | HIVE               | 0.0527                    | 0.8567                  | 0.7855         | 0.1978         | -2.07        | 0.12                  |
> | MagicBrush         | 0.1011                    | 0.8526                  | **0.8278**     | **0.2516**     | -2.13        | 0.11                  |
> | HQ-Edit (Ours)     | **0.1067**                | **0.8588**              | 0.8139         | 0.2231         | **-1.97**    | **0.13**              |
>
> As shown in the table, HQ-Edit performs the best in terms of CLIP_dir, CLIP_img, and Reward Score (Jiazheng et al. [1]) and ranks second-best in terms of DINO and SSIM metrics. Some qualitative results are shown in the `real_image_results.png` in this [anonymous repo](https://anonymous.4open.science/r/hqedit_re-8D01). With these results, We conclude that the superb quality and large size of HQ-Edit can still enable models to handle real-life and photorealistic images reasonably well.
>
> Moreover, we believe this domain issue will be addressed in the future. For example, one possible strategy is to design a (sophisticated) mixture training algorithm to mitigate this distribution bias. This issue may also be effectively alleviated by switching from DALL-E 3 to more recent T2I models like FLUX, which can generate very photorealistic images to construct an upgraded version of HQ-Edit.
>
> We will add these discussions in the next version.
>
> Please see **Common Concerns/Performance on real images (EMU Edit)** for more.
>
>
> ## Using more popular metrics
>
> Thanks. Following your suggestion, we first present evaluation results with more classic metrics, including the most commonly used CLIP directional similarity ($\text{CLIP}_\text{dir}$), CLIP image similarity ($\text{CLIP}_\text{img}$), DINO image similarity (DINO), and SSIM. The results are as follows:
>
> | Model       | $\text{CLIP}_\text{dir}$       | $\text{CLIP}_\text{img}$       | DINO       | SSIM           |
> |--------------|-----------------|------------------|----------------|----------------|
> | InstructPix2Pix         | 0.0444 | 0.7353 | 0.7252 | 0.1673 |
>  |HIVE   | 0.0970 | 0.8633 | 0.8851 |  0.4646 |
> | MagicBrush   | 0.1109 | 0.8173 | 0.8246 | 0.2735 |
> | HQ-Edit (Ours)         | **0.1351** | **0.9246** | **0.9692** | **0.6561** |
>
> We can observe that the model trained with HQ-Edit consistently yields the highest performance across all metrics.
>
> Please see **Common Concerns/Other evaluation metrics and human evaluation** for more.
>
>
> ## Human evaluation for Coherence
> Thanks for raising this concern. We conducted a human evaluation with 160 image pairs formed with output images from HQ-Edit. Human evaluators are tasked with determining which image in each pair is better. "Image 1 is better", "Tie" and "Image 2 is better" are encoded as -1, 0 and 1. We then compute the Pearson Correlation between human evaluation scores and deltas between the GPT-evaluated scores of the first and second images.
>
> We also employ the ImageReward (Jiazheng et al. [1]), which developed a BLIP-based model for evaluating alignment between text-to-image pairs and human preference, as the reward model for evaluating the output image quality. The correlation between scores from ImageReward and human evaluation is computed in the same way. The results are as follows:
> | Coherence | Reward Model | Reward Model (after Sigmoid) |
> | - | - | - |
> | 0.27 | 0.12 | 0.10 |
>
> The results above suggest that the Coherence score we proposed is significantly more aligned with human preference than the newly proposed reward model, ImageReward.
>
> Please see **Common Concerns/Other evaluation metrics and human evaluation** for more.

---

> > ### Author Response · Authors · 2024-11-25
> >
> > ## Can GPT-4V truly differentiate between scores that are a few points apart?
> > Thanks for raising this interesting question. We calculate the per-image deltas of Alignment and Coherence between InstructPix2Pix, HIVE, MagicBrush, and HQ-Edit. The averaged per-image **absolute** deltas are shown below.
> >
> > |  Alignment | InstructPix2Pix   | HIVE   | MagicBrush | HQ-Edit     |
> > |-----------|--------|--------|------------|--------|
> > | **InstructPix2Pix**  | 0.00   | 12.56  | 14.27      | 12.51  |
> > | **HIVE**  | 12.56  | 0.00   | 9.87       | 9.21   |
> > | **MagicBrush** | 14.27 | 9.87  | 0.00       | 10.51  |
> > | **HQ-Edit**    | 12.51  | 9.21   | 10.51      | 0.00   |
> >
> > |  Coherence | InstructPix2Pix   | HIVE   | MagicBrush | HQ-Edit     |
> > |-----------|--------|--------|------------|--------|
> > | **InstructPix2Pix**  | 0.00   | 27.44  | 38.72      | 25.99  |
> > | **HIVE**  | 27.44  | 0.00   | 32.58      | 26.02  |
> > | **MagicBrush** | 38.72 | 32.58  | 0.00       | 36.31  |
> > | **HQ-Edit**    | 25.99  | 26.02  | 36.31      | 0.00   |
> >
> > We can observe that the per-image deltas of Alignment scores are around 10 and Coherence scores around 30, meaning metrics for each image are differentiated by a large margin. Moreover, as provided in the **common responses**, our GPT metrics are consistent with other metrics (like different CLIP scores, DINO scores, ImageReward scores [1], and human evals).
> >
> > Given this evidence, we believe our GPT metrics are valid and can accurately tell the quality differences between images.
> >
> > [1] Xu, J., et al. (2023). ImageReward: Learning and Evaluating Human Preferences for Text-to-Image Generation. In NeurIPS 2023

---

> ### Comment · Reviewer_Drub · 2024-11-26
>
> Thank you for your response~

---

> > ### Author Response · Authors · 2024-12-02
> > **followup**
> >
> > Dear Reviewer Drub
> >
> > As the discussion deadline is approaching, we would like to check if our additional responses (https://openreview.net/forum?id=mZptYYttFj&noteId=UVE8tLBjAJ) address/mitigate your concerns.
> >
> > Looking forward to your response, and we are happy to provide more info if needed.
> > Thanks, Authors of 8390

---

### Official Review · Reviewer_PppU · 2024-11-04

**Soundness:** 3
**Presentation:** 3
**Contribution:** 3
**Rating:** 6
**Confidence:** 4

**Summary:**

This paper introduces HQ-Edit for instruction-based image editing task, which contains around 200,000 edits. By using GPT-4V and DALL-E 3, the author develops a scalable data collection pipeline, which consists of three steps (Expansion, Generation and Post-Processing). Besides, the author also designs two metrics by using GPT-4V, which own a higher correlation to human preference compared to the CLIP score. Based on HQ-Edit, the finetuned InstructPix2Pix can attain state-of-the-art image editing performance.

**Strengths:**

1. The proposed HQ-Edit can be a training data for instruction-based image editing task, which can promote the development of this area.
2. The performance of finetuned InstructPix2Pix has proven the effectiveness of HQ-Edit.
3. The proposed evaluation metrics are superior to the CLIP score.

**Weaknesses:**

1. It seems that HQ-Edit only contains non-rigid pair data.
2. There are many types of operations for instruction-based image editing task (e.g., object addition, object removal, non-rigid operation, local transformation, global transformation). The figure 8 and figure 9 only show the transformation part. The author should show the results of all these operations to make a comprehensive comparison.
3. Regarding metrics, since there is already a large amount of Human Evaluation Scores, why not use a model to learn a reward model for scoring? Alternatively, we could fine-tune an MLLM for scoring. Simply using prompt engineering with GPT-4V may not be the best option.
4. There is some typo in line 512 – 516. It should be “enhances its performance on Coherence but hurts alignment”.
5. What is the architecture of InstructPix2Pix trained by the author? (Base or XL). If the XL version is finetuned with HQ-Edit, can the performance be further improved than Base version?

**Questions:**

Please refer to the weaknesses part.

---

> ### Author Response · Authors · 2024-11-25
>
> We thank the reviewer for the detailed comments and agree that the empirical evidence supports HQ-Edit's effectiveness as a finetuning dataset, and the proposed GPT-based evaluation is superior to the currently popular metric, which is the CLIP score. We address the concerns as follows:
>
> ## HQ-Edit only contains non-rigid pair data
> Sorry for the confusion. We would like to clarify that our HQ-Edit actually includes rigid operations like changing object viewpoint and rotating objects, e.g., viewpoint alteration, as shown in Fig. 1(a); object replacement, as shown in Fig. 1(d) and Fig. 10(left); object addition, as shown in Fig. 9(rightmost image).
>
> More examples are available in this [anonymous repo](https://anonymous.4open.science/r/hqedit_re-8D01). See `rigid_data_examples.png` for more samples for objection addition, object removal, object replacement, and viewpoint alteration.
>
> ## More qualitative results with rigid operation
> Thanks for the suggestion. We just add such examples, which include operations such as "add objects", "remove objects", "replace objects", available at the `rigid_operation_results.png` in this [anonymous repo](https://anonymous.4open.science/r/hqedit_re-8D01)., to illustrate the corresponding visual comparison. It is shown that the model finetuned with HQ-Edit yields generally better results in terms of the instruction following and image quality compared to InstructPix2Pix, MagicBrush, and HIVE.
>
> ## Reward model for evaluation
> Thanks for the suggestion. We employ ImageReward (Jiazheng et al. [1]), which develops a BLIP-based reward model for accessing text-to-image pairs' alignment with human preference, as the reward model for measuring the quality of output images. Since scores from ImageReward are unbounded, we normalize reward scores to $[0, 1]$ with Sigmoid. The results are as follows:
>
> | Model       | Reward Score | Reward Score (after Sigmoid)|
> |--------------|-----------------|------------------|
> | InstructPix2Pix         | -0.54|0.40|
> |HIVE| -0.09|0.48|
> | MagicBrush   | -0.05|0.49|
> | HQ-Edit (Ours)         | **-0.03**| **0.50** |
>
> One can see that the model trained with HQ-Edit yields the highest performance.
>
> We also present the human evaluation results with Coherence and ImageReward scores. We conducted a human evaluation with 160 image pairs formed with output images from HQ-Edit. Human evaluators are tasked with determining which image in each pair is better. "Image 1 is better", "Tie" and "Image 2 is better" are encoded as -1, 0 and 1. We then compute the Pearson Correlation between human evaluation scores and deltas between the GPT-evaluated scores of the first and second images. The correlation between scores from ImageReward and human evaluation is computed in the same way.
>
> | Coherence | Reward Model | Reward Model (after Sigmoid) |
> | - | - | - |
> | 0.27 | 0.12 | 0.10 |
>
> The results above suggest that the Coherence score we proposed is significantly more aligned with human preference than the newly proposed reward model. As the development of reward models for image generation and image editing is still at a preliminary stage, we find, alternatively, using a mature and well-tested general-purpose MLLM that is GPT-4V is valid and reasonable (though maybe not the best) at this stage.
>
> Please see **Common Concerns/Other evaluation metrics and human evaluation** for more.
>
> ## Typo in Sec4.4
> Thanks for pointing this out. We will fix this in the revision.
>
> ## Finetuned architecture of InstructPix2Pix
> Sorry for missing the details. The model we finetuned is the Base version. We expect the switch to XL will lead to further performance improvement because of the large number of high-quality training samples provided by HQ-Edit, but we are unable to empirically verify it during the rebuttal stage mainly due to computing resource limitation (e.g., compared to Base, training XL requires much larger GPU memory consumption and much longer training time).
>
>
> [1] Xu, J., et al. (2023). ImageReward: Learning and Evaluating Human Preferences for Text-to-Image Generation. In NeurIPS 2023

---

> > ### Comment · Reviewer_PppU · 2024-11-28
> > **Official Comment by Reviewer PppU**
> >
> > Thank you for the detailed response. I will increase my rating.

---

> > > ### Author Response · Authors · 2024-12-02
> > > **followup**
> > >
> > > Dear Reviewer PppU
> > >
> > > As the discussion deadline is approaching, we would like to check if our additional responses (https://openreview.net/forum?id=mZptYYttFj&noteId=UVE8tLBjAJ) address/mitigate your concerns.
> > >
> > > Looking forward to your response, and we are happy to provide more info if needed.
> > > Thanks, Authors of 8390

---

### Official Review · Reviewer_ptsc · 2024-11-04

**Soundness:** 3
**Presentation:** 3
**Contribution:** 3
**Rating:** 8
**Confidence:** 4

**Summary:**

This paper introduces a large-scale and high quality instruction-based image editing dataset, HQ-Edit, with 200K edits. This dataset contributes to address the challenges of low instruction alignment and quality in image editing by employing GPT-4, DALL-E3 and GPT-4V. This work presents a new and scalable data curation process involving with crafted prompt engineering and diptych generation. The authors also present baselines finetuned on HQ-Edit which shows state-of-the-art performance in instruction-based single image editing task.

**Strengths:**

1. Compared with previous work, the proposed dataset is high quality regarding to image resolution, content/edit-type diversity and prompt-image alignment.
2. Baseline method InstructPix2Pix finetuned on the proposed HQ-Edit achieves state-of-the-art performance.
3. The proposed data curation pipeline is scalable by leveraging pretrained generative model (e.g. DALL·E3) and visual-language model (e.g. GPT-4 / GPT-4V).
4. The paper is well-organized and easy to follow.

**Weaknesses:**

1. Compared with previous work, e.g. MagicBrush, the source images from HQ-Edit are generated by DALLE-3, which may introduce distribution bias between AIGC and photo realistic contents.
2. The necessity/importance analysis of using diptych generation is missing.
3. The proposed two metrics Alignment and Coherence are mainly used in the main evaluation. Given the validated limitation of CLIP directional similarity, other commonly used metrics are missing for quantitative evaluation, which may lead to unfair comparison.

**Questions:**

Referring to weakness 1, can you provide an analysis or design an experiment to show the potential generalization ability of the finetuned baseline model on photo-realistic data?

---

> ### Author Response · Authors · 2024-11-25
>
> We thank the reviewer for the detailed comments and find that the HQ-Edit dataset poccess high quality and utility, the data collection pipeline is scalable and well-defined, and the final model's performance is impressive. We address your concerns as follows:
>
> ## Distribution shifting with synthetic images
> Thank you for raising this concern. Yes, this is one limitation of our work at this stage. To better understand how our model handles real images, we first evaluate on the test set of EMU Edit, which consists only of real-life images.
> | Model          | $\text{CLIP}_\text{dir}$ | $\text{CLIP}_\text{img}$ | DINO        | SSIM           | Reward Score | Reward Score (after Sigmoid) |
> |---------------|---------------------------|--------------------------|----------------|----------------|--------------|-----------------------------|
> | InstructPix2Pix    | 0.0775                    | 0.8396                  | 0.7879         | 0.2223         | -2.05        | 0.12                  |
> | HIVE               | 0.0527                    | 0.8567                  | 0.7855         | 0.1978         | -2.07        | 0.12                  |
> | MagicBrush         | 0.1011                    | 0.8526                  | **0.8278**     | **0.2516**     | -2.13        | 0.11                  |
> | HQ-Edit (Ours)     | **0.1067**                | **0.8588**              | 0.8139         | 0.2231         | **-1.97**    | **0.13**              |
>
> As shown in the table, HQ-Edit performs the best in terms of CLIP_dir, CLIP_img, and Reward Score (Jiazheng et al. [1]) and ranks second-best in terms of DINO and SSIM metrics. Some qualitative results are shown in the `real_image_results.png` in this [anonymous repo](https://anonymous.4open.science/r/hqedit_re-8D01). With these results, We conclude that the superb quality and large size of HQ-Edit can still enable models to handle real-life and photorealistic images reasonably well.
>
> Moreover, we believe this domain issue will be addressed in the future. For example, one possible strategy is to design a (sophisticated) mixture training algorithm to mitigate this distribution bias. This issue may also be effectively alleviated by switching from DALL-E 3 to more recent T2I models like FLUX, which can generate very photorealistic images to construct an upgraded version of HQ-Edit.
>
> We will add these discussions in the next version.
>
> Please see **Common Concerns/Performance on real images (EMU Edit)** for more.
>
> ## Necessity of diptych generation
> Thank you for raising this concern. As detailedly explained in the **Common Concerns**, we use diptych generation mainly as a workaround so that DALL-E 3, whose API does not innately support image-to-image generation, can be used to generate image pairs that have high relevance. We will add these clarifications in the next version.
>
> Furthermore, to accurately and effectively extract info from generated diptych, we proposed a series of post-processing methods, including:
> 1. Image-wise
>    We further improved image alignment using **Warping & Filtering**. As validated empirically in Sec. 4.4 and visually demonstrated in Fig. 3, these designs can improve image quality and, consequently, the model's performance, enabling our proposed dataset to further enhance the image editing models' capabilities, such as InstructPix2Pix.
> 2. Text-wise
>    Additionally, we leveraged **Instruction Refinement**, using GPT-4V to refine the original editing instructions further, ensuring all still-existing misalignments after image post-processing are encompassed in editing instructions. Arguably, this makes all the image pairs achieve almost precise alignment as they correspond to the rewritten instructions in detail.
>
> Please see **Common Concerns/Necessity of diptych generation** for more.
>
>
> ## More classic metrics in evaluation
> Thanks. Following your suggestion, we first present evaluation results with more classic metrics, including the most commonly used CLIP directional similarity ($\text{CLIP}_\text{dir}$), CLIP image similarity ($\text{CLIP}_\text{img}$), DINO image similarity (DINO), and SSIM. The results are as follows:
>
> | Model       | $\text{CLIP}_\text{dir}$       | $\text{CLIP}_\text{img}$       | DINO       | SSIM           |
> |--------------|-----------------|------------------|----------------|----------------|
> | InstructPix2Pix         | 0.0444 | 0.7353 | 0.7252 | 0.1673 |
>  |HIVE   | 0.0970 | 0.8633 | 0.8851 |  0.4646 |
> | MagicBrush   | 0.1109 | 0.8173 | 0.8246 | 0.2735 |-2.13|0.11
> | HQ-Edit (Ours)         | **0.1351** | **0.9246** | **0.9692** | **0.6561** |
>
> We can observe that the model trained with HQ-Edit consistently yields the highest performance across all metrics.
>
> Please see **Common Concerns/Other evaluation metrics and human evaluation** for more.
>
>
> [1] Xu, J., et al. (2023). ImageReward: Learning and Evaluating Human Preferences for Text-to-Image Generation. In NeurIPS 2023

---

> > ### Author Response · Authors · 2024-12-02
> > **rebuttal**
> >
> > Dear Reviewer ptsc
> >
> > Thanks again for strongly supporting our work. Could you please let us know if our rebuttal successfully addresses/mitigates your concerns?
> >
> > Thanks
> > Authors of 8390

---

### Author Response · Authors · 2024-11-25
**Common Concerns**

We thank all reviewers for their thoughtful feedback, which will help us improve the quality of this paper. We are delighted to see all the reviewers agree that the model finetuned with our HQ-Edit achieved strong performance, indicating the quality and utility of HQ-Edit and, hence, our data pipeline's effectiveness. Additionally, reviewers acknowledge that the data collection pipeline is innovative, scalable, and well-defined (k3ov, U5Yr), and the metrics we proposed align well with human evaluation and can be used for quantitatively assess the editing quality (cVQk, U5Yr), and contains a variety of diverse editing operations, showing its versatility (Drub)

We first address some common concerns as follows:

## Necessity of diptych generation
We would like to clarify that this design choice is mainly driven by the fact that DALL-E 3 doesn't have a built-in editing API --- i.e., it **cannot generate an image given an input image**. This also marks **one of the key differences** between our pipeline and others, such as InstructPix2Pix. Given this restriction, we alternatively utilize diptych generation as a workaround so that it is possible to leverage DALL-E 3's powerful text-to-image generation ability.

Furthermore, to accurately and effectively extract info from the generated diptych, we proposed a series of post-processing methods, including:
1. Image-wise
   We further improved image alignment using **Warping & Filtering**. As validated empirically in Sec. 4.4 and visually demonstrated in Fig. 3, these designs can improve image quality and, consequently, the model's performance, enabling our proposed dataset to further enhance the image editing models' capabilities, such as InstructPix2Pix.
2. Text-wise
   Additionally, we leveraged **Instruction Refinement**, using GPT-4V to refine the original editing instructions further, ensuring all still-existing misalignment after image post-processing are encompassed in editing instructions. Arguably, this makes all the image pairs achieve almost precise alignment as they correspond to the rewritten instructions in detail.

## Other evaluation metrics and human evaluation
Some reviewers are concerned with the two GPT-based metrics we proposed and find results lack classic metrics.

To address this concern, we first present evaluation results with more metrics, including the most commonly used CLIP directional similarity (CLIP_dir), CLIP image similarity (CLIP_img), DINO image similarity (DINO), and SSIM. Also, we employ ImageReward (Jiazheng et al. [1]) as the reward model for measuring the quality of output images, according to the suggestion from Reviewer PppU. Since scores from ImageReward are unbounded, we normalize reward scores to $[0, 1]$ with Sigmoid. The results are as follows:

| Model       | $\text{CLIP}_\text{dir}$       | $\text{CLIP}_\text{img}$       | DINO       | SSIM           | Reward Score | Reward Score (after Sigmoid)
|--------------|-----------------|------------------|----------------|----------------|----------------|----------------|
| InstructPix2Pix         | 0.0444 | 0.7353 | 0.7252 | 0.1673 | -0.54|0.40|
 |HIVE   | 0.0970 | 0.8633 | 0.8851 |  0.4646 | -0.09|0.48|
| MagicBrush   | 0.1109 | 0.8173 | 0.8246 | 0.2735 | -0.05|0.49|
| HQ-Edit (Ours)         | **0.1351** | **0.9246** | **0.9692** | **0.6561** | **-0.03**| **0.50** |

We can observe that the model trained with HQ-Edit consistently yields the highest performance across all metrics, further confirming its effectiveness.

Additionally, we present the human evaluation results with Coherence and ImageReward scores. Specifically, we conducted a human evaluation with 160 image pairs formed with output images from HQ-Edit. Human evaluators are tasked with determining which image in each pair is better. "Image 1 is better", "Tie" and "Image 2 is better" are encoded as -1, 0 and 1. We then compute the Pearson Correlation between human evaluation scores and deltas between the GPT-evaluated scores of the first and second images. The correlation between scores from ImageReward and human evaluation is computed in the same way.

| Coherence | Reward Model | Reward Model (after Sigmoid) |
| - | - | - |
| 0.27 | 0.12 | 0.10 |

The results above suggest that the Coherence score we proposed is significantly more aligned with human preference than the newly proposed reward model. As the development of reward models for image generation and image editing is still at a preliminary stage, we find alternatively, using a mature and well-tested general-purpose MLLM that is GPT-4V is valid and reasonable at this stage.

We will add these additional results and corresponding discussions in the next version.

---

> ### Author Response · Authors · 2024-11-25
>
> ## Performance on real images (EMU Edit)
>
> We first systematically evaluate our model on the test set of EMU Edit, which consists only with real-life images.
> | Model          | $\text{CLIP}_\text{dir}$ | $\text{CLIP}_\text{img}$ | DINO        | SSIM           | Reward Score | Reward Score (after Sigmoid) |
> |---------------|---------------------------|--------------------------|----------------|----------------|--------------|-----------------------------|
> | InstructPix2Pix    | 0.0775                    | 0.8396                  | 0.7879         | 0.2223         | -2.05        | 0.12                  |
> | HIVE               | 0.0527                    | 0.8567                  | 0.7855         | 0.1978         | -2.07        | 0.12                  |
> | MagicBrush         | 0.1011                    | 0.8526                  | **0.8278**     | **0.2516**     | -2.13        | 0.11                  |
> | HQ-Edit (Ours)     | **0.1067**                | **0.8588**              | 0.8139         | 0.2231         | **-1.97**    | **0.13**              |
>
> As shown in the table, HQ-Edit performs the best in terms of CLIP_dir, CLIP_img, and Reward Score (Jiazheng et al. [1]) and ranks second-best in terms of DINO and SSIM metrics. Some qualitative results are provided in the `real_image_results.png` in this [anonymous repo](https://anonymous.4open.science/r/hqedit_re-8D01). With these results, we can conclude that the superb quality and large size of HQ-Edit can still enable models to reasonably handle real-life and photorealistic images despite the domain gap. We also believe that adapting our framework to collect data from the more recent T2I models like FLUX--which can generate more photorealistic images--will enable us to see stronger performance on EMU Edit.
>
> Lastly, compared to collecting real images, we stress that using synthetic images can be more scalable and comprehensive. We believe that a stronger editing dataset could be a mixture of synthetic and real images, where synthetic images contribute to comprehensiveness and precision—properties that may be challenging to achieve with real images alone.
>
>
> [1] Xu, J., et al. (2023). ImageReward: Learning and Evaluating Human Preferences for Text-to-Image Generation. In NeurIPS 2023

---

### Author Response · Authors · 2024-12-01
**A General Response Regarding Reviewer x6Ut's Concerns**

Dear All Reviewers,

We appreciate the thoughtful feedback provided by all reviewers and would like to address the concerns raised by Reviewer x6Ut, which may be shared by others. Our response comprises two parts: (1) presenting a practical and successful example of utilizing image generation models for image editing, and (2) reiterating the value of our HQEdit submission.


1.  **A Successful Example of Utilizing Image Generation for Image Editing**
We are pleased to share a direct commercial follow-up of HQEdit that incorporates significant scaling and further enhancements. We have updated the results of this commercial model in [real_image_results.png](https://anonymous.4open.science/r/hqedit_re-8D01/real_image_results.png) and [rigid_operation_results.png](https://anonymous.4open.science/r/hqedit_re-8D01/rigid_operation_results.png). These results demonstrate that the previously concerning errors—such as background alteration or unintended transformations—can be substantially mitigated.
Regarding technique enhancement, this commercial model makes three main changes: 1) it generates a large-scale pairwise dataset with more randomness to ensure diversity, and then we apply filters to choose good examples for model training; 2) in the model training, a causal self-attention structure is introduced for enhancing image conditioning; 3) based on the current state of the editing model, it prepares a new set of data following a similar generation pipeline; and the results are labeled, filtered, and used to fine-tune the editing model. This process is repeated multiple times until convergence. We hope this strong result offered by our commercialized model can alleviate reviewers' concerns about the practical viability of using image generation models to produce training data for successful image editing models in real-world applications.

2) In this second part, we would like to invite all reviewers to **re-assess the valve of this ICLR submission**. Specifically, As one of the pioneering works in this field, we believe our HQEdit alone has already provided substantial evidence that utilizing image generation models for image editing is a promising direction. Our paper and this rebuttal offer comprehensive evaluations showing that **HQEdit achieves superior performance compared to other baselines**, measured by different classic metrics (like CLIP scores), our proposed GPT scores (alignment and coherence), and human-valve-based reward model scores.  Furthermore, our pipeline is highly scalable for generating vast amounts of synthetic images for training purposes. We believe this scalability is crucial for advancing future research in image editing. As history has shown in deep learning, large quantities of somewhat noisy data can significantly boost model performance compared to small datasets with extremely high accuracy. We believe this principle is equally applicable to image editing, thereby potentially underscoring the importance and research value of our HQEdit approach.

We are grateful for your consideration and are happy to address any further questions or concerns you may have.

Thanks
Authors

---

### Meta-Review · Area_Chair_S64c · 2024-12-21

**Metareview:**

The paper presents an approach for generating a large-scale synthetic (generated) dataset for training models to perform instruction-based image editing (similar to InstructPix2Pix). Unlike InstructPix2Pix, which uses an open-source text-to-image model to generate pairs of corresponding images via attention-hacking, this approach instead uses a closed-source image generator as a black box (Dall-E3), followed by a handful of filtering operations driven by vision-language models. The resulting dataset can be used to fine-tune editing models, resulting in SoTA performance.

Reviewers acknowledge the value of the dataset, and that evaluations show improved performance over the state-of-the-art. Some reviewers question the utility of a synthetic dataset for training models to operate on real data, as well as the particular methodology of the way in which the closed-source model was used (i.e., generating image pairs jointly as a diptych).

Ultimately, the review remain split, with a spread across all different scores. Although the proposed method may not be the ideal solution for training the top-quality commercial models, the approach used to filter the data may have interesting insights for future work, and the current evaluations are sufficiently convincing to validate that it provides an improvement over prior work. I advocate for the paper's acceptance.

**Additional Comments On Reviewer Discussion:**

Reviews were initially generally positive, except for one overwhelmingly negative review from x6Ut, which swayed PPpu to lower their score. A long discussion ensued questioning the value of a large synthetic dataset, and whether it would be useful for training commercial-quality editing models.

Ultimately, the review remain split, with a spread across all different scores. This AC views the strong negative review as unnecessarily harsh---even if the proposed method may not lead to the best quality commercial models, the approach used to filter the data may have interesting insights for future work, and the current evaluations are sufficiently convincing to validate that it provides an improvement over prior work.

---

### Decision · Program_Chairs · 2025-01-22

Accept (Poster)